# An augmented Mendelian randomization approach provides causality of brain imaging features on complex traits in a single biobank-scale dataset

Anyi Yang [1,2], Yucheng T. Yang [1,2]*, Xing-Ming Zhao [1,2,3,4]*

1 Department of Neurology, Zhongshan Hospital and Institute of Science and Technology for Brain-Inspired Intelligence, Fudan University, Shanghai, People's Republic of China, 2 MOE Key Laboratory of Computational Neuroscience and Brain-Inspired Intelligence, and MOE Frontiers Center for Brain Science, Fudan University, Shanghai, People's Republic of China, 3 State Key Laboratory of Medical Neurobiology, Institutes of Brain Science, Fudan University, Shanghai, People's Republic of China, 4 International Human Phenome Institutes (Shanghai), Shanghai, People's Republic of China

* yangyy@fudan.edu.cn (YTY); xmzhao@fudan.edu.cn (XMZ)

**Data Availability Statement:** Genotype, neuroimaging and complex traits data used in the present study is available from the UK Biobank with restrictions applied. Data were used under license

## Abstract

Mendelian randomization (MR) is an effective approach for revealing causal risk factors that underpin complex traits and diseases. While MR has been more widely applied under two-sample settings, it is more promising to be used in one single large cohort given the rise of biobank-scale datasets that simultaneously contain genotype data, brain imaging data, and matched complex traits from the same individual. However, most existing multivariable MR methods have been developed for two-sample setting or a small number of exposures. In this study, we introduce a one-sample multivariable MR method based on partial least squares and Lasso regression (MR-PL). MR-PL is capable of considering the correlation among exposures (e.g., brain imaging features) when the number of exposures is extremely upscaled, while also correcting for winner's curse bias. We performed extensive and systematic simulations, and demonstrated the robustness and reliability of our method. Comprehensive simulations confirmed that MR-PL can generate more precise causal estimates with lower false positive rates than alternative approaches. Finally, we applied MR-PL to the datasets from UK Biobank to reveal the causal effects of 36 white matter tracts on 180 complex traits, and showed putative white matter tracts that are implicated in smoking, blood vascular function-related traits, and eating behaviors.

## Author summary

Mendelian randomization (MR) can be a powerful tool for uncovering causal risk factors that underpin complex traits and diseases. MR framework has been successfully applied in understanding casual relationships between brain imaging and complex traits. However, most of these studies rely on two different samples and univariable MR framework, leading to the issues such as sample dis-harmonization and ignorance of

and thus are not publicly available. To obtain access to the UK Biobank data, a standard protocol can be followed at https://www.ukbiobank.ac.uk/register-apply/. The GWAS summary statistics utilized in this study is publicly available at https://github.com/BIG-S2/GWAS. The previous reported GWAS associations listed in the NHGRI-EBI GWAS catalog can be directly downloaded from https://www.ebi.ac.uk/gwas/docs/file-downloads. The codes used to perform MR-PL and reproduce the results in this study are publicly available on GitHub at https://github.com/ZhaoXM-Lab/MR-PL.

**Funding:** XMZ was supported by National Key Research and Development Program of China (Grant No. 2023YFF1204800), National Natural Science Foundation of China (Grant No. T2225015, 61932008), Shanghai Municipal Science and Technology Major Project (Grant No. 2018SHZDZX01), and Greater Bay Area Institute of Precision Medicine (Guangzhou) (Grant No. IPM21C008). YTY was supported by National Key Research and Development Program of China (Grant No. 2021YFF0703703), and Natural Science Foundation of Shanghai (Grant No. 21ZR1408100). The funders had no role in study design, data collection and analysis, decision to publish, or preparation of the manuscript.

**Competing interests:** The authors have declared that no competing interests exist.

the high correlation between different brain regions. To overcome these limitations, we develop a new one-sample multivariable MR approach named MR-PL, which has a uniform statistical framework and better performance. Briefly, MR-PL considers the complex correlations among numerous risk factors and provides a flexible step for the bias correction. MR-PL can perform efficient phenome-wide scan for potential causality between a wide spectrum of brain imaging features and complex traits on a biobank-scale dataset with a scale of 10,000's individual-level data. We demonstrate the power of our method by studying the causal relationships between 36 white matter tracts and 180 complex traits using the datasets from UK Biobank. We show that certain white matter tracts could be causally linked to smoking, blood vascular function, and eating behaviors.

## Introduction

Mendelian randomization (MR) serves as an alternative for coping with the challenge of identifying causality rather than randomized controlled trials (RCT), the gold standard for causal inference but generally impractical and unethical. MR employs analytic methods to probe the causal effects of an exposure variable on an outcome variable based on genetic variants. In general, effective MR methods conform to three explicit assumptions as follows [1]: (i) Relevance: the genetic variant is associated with the exposure; (ii) Independence: each genetic variant is not associated with confounders; (iii) Exclusion restriction: each genetic variant affects the outcome only through the exposure. Existing research has suggested that genetic risk factors could affect complex diseases by acting on endophenotypes of brain traits [2]. Accordingly, MR has been a vital workflow for understanding the causality between the brain and complex neurological, psychological, and behavioral traits. More recently, there have been dramatic advancement in the field of brain imaging technologies, and genome-wide association studies (GWASs) have been widely conducted on brain imaging and disorders. Under this context, the MR studies have been applied to detect the putative causal relationships between brain imaging-derived phenotypes (IDPs) and complex traits or disorders [3–6]. For instance, a recent study investigated the causal relationships between 587 IDPs (e.g., cortical volume, area, thickness and white matter microstructure) and 10 psychiatric disorders and reported 11 causal pairs (e.g., the causal association of right superior fronto-occipital fasciculus fractional anisotropy and left accumbens volume on schizophrenia) [3]. Another study examined the causal relationships between 110 diffusion tensor imaging (DTI) measurements and 11 psychiatric disorders and revealed that the superior longitudinal fasciculus degeneration may serve as a risk factor for anorexia nervosa [4].

There have been several challenges in the investigation of the causal effects of IDPs on the outcome. First, most of the above-mentioned studies examining IDP-trait links are based on two-sample MR, where genetic associations of the exposure and outcome were examined in two independent cohorts with GWAS summary statistics. In general, two-sample MR benefits from the power of larger GWAS sample size, whereas it suffers from many concerns affecting the causal estimate results [7]: (i) dis-harmonization of population characteristics and processing pipelines between the two samples; (ii) limitations in the analyses as only summary statistics datasets are accessible; (iii) overlapping participants between the two cohorts as several large consortia tend to have the identical individuals. Moreover, biobank-scale datasets with well-collected brain imaging data, genotype data and other detailed phenotype information are increasingly available, such as ABCD (http://www.nih.gov/about/disclaim.html), UK

Biobank (UKBB, http://www.ukbiobank.ac.uk/), CHARGE (https://www.hgsc.bcm.edu/human/charge-consortium), and ENIGMA (http://enigma.ini.usc.edu/). On that basis, the phenome-wide scan of potential causality between a wide spectrum of brain IDPs and other complex health-associated traits can be conducted on the scale of 10,000's or 1,000's individual-level data.

Second, a univariable MR framework has been constantly employed in the above-described studies examining IDP-trait links. However, brain functions rely on effective communications across different regions, and highly structural, functional, and genetic correlation have been reported between different brain regions [8–10]. Thus, a genetic variant is likely to affect the outcome through multiple brain regions mutually correlated with each other. Since canonical MR cannot allow variants to have pleiotropic effects on the outcome except through the exposure, it will generate misleading inferences when only one single imaging feature serves as exposure. In contrast to univariable MR, multivariable MR is capable of more effectively coping with the pleiotropy problem, thus allowing the genetic variants related to multiple exposures [11]. Existing multivariable MR methods are generally designed on two-sample setting [11–16], or a small number of exposures [16–19]. While the method MVMR-cML can account for sample-overlapping within the two-sample MR framework, its applicability is limited to a small number of exposures [16]. Another interesting work has proposed a MR framework to jointly select multiple instrumental variables and exposures [20]. This framework is dependent on a transformation of exposures into a single synthetic exposure based on dimensionality reduction techniques, such that the causal effect of each individual exposure on the outcome cannot be revealed. Moreover, a considerable amount of MR studies on neuroimaging data selected instruments from the identical dataset employed for analyses [3,21]. However, this can bias MR estimates and trigger "winner's curse", suggesting that genetic associations tend to have upward bias in the dataset when initially identified [7]. The above-mentioned problem is more serious for multivariable MR. To be specific, due to the diversity of brain templates and IDPs (i.e., subcortical, cortical, white matter) and the population from the biobank (i.e., UKBB), it is difficult to make instrument selection on a totally independent dataset. Overall, more effective means should be implemented in one-sample multivariable MR setting with multiple imaging features as exposures.

Here, we propose MR-PL, a one-sample multivariable MR method, which enables causal effect estimation from multiple exposures from one single biobank-based dataset. MR-PL considers the correlation among exposures and corrects the winner's curse bias. The simulation results suggested that MR-PL outperformed alternative approaches, standing out for the power to make precise causal effect estimation and control type I error rate. To verify the utility of MR-PL, the proposed method was applied to estimate the causal effects of 36 white matter tracts on 180 complex traits using data from UKBB, with most of the predicted causal relationships can be supported by prior knowledge. In general, the proposed method is effective in revealing the potentially causal relationships between a set of exposures (e.g., IDPs) and massive complex traits collected from a single biobank-scale cohort.

## Results

### Overview of MR-PL

MR-PL relies on a one-sample multivariable MR framework, with the aim of revealing the causal effects of multiple correlated exposures (variables assumed as the risk factors) on the outcome (another variable, usually serving as a disorder or trait) using individual-level data from a single biobank-based dataset. MR-PL is a universal method that can be applied to diverse high-dimensional omics data (e.g., transcriptome, proteome, metabolome, and

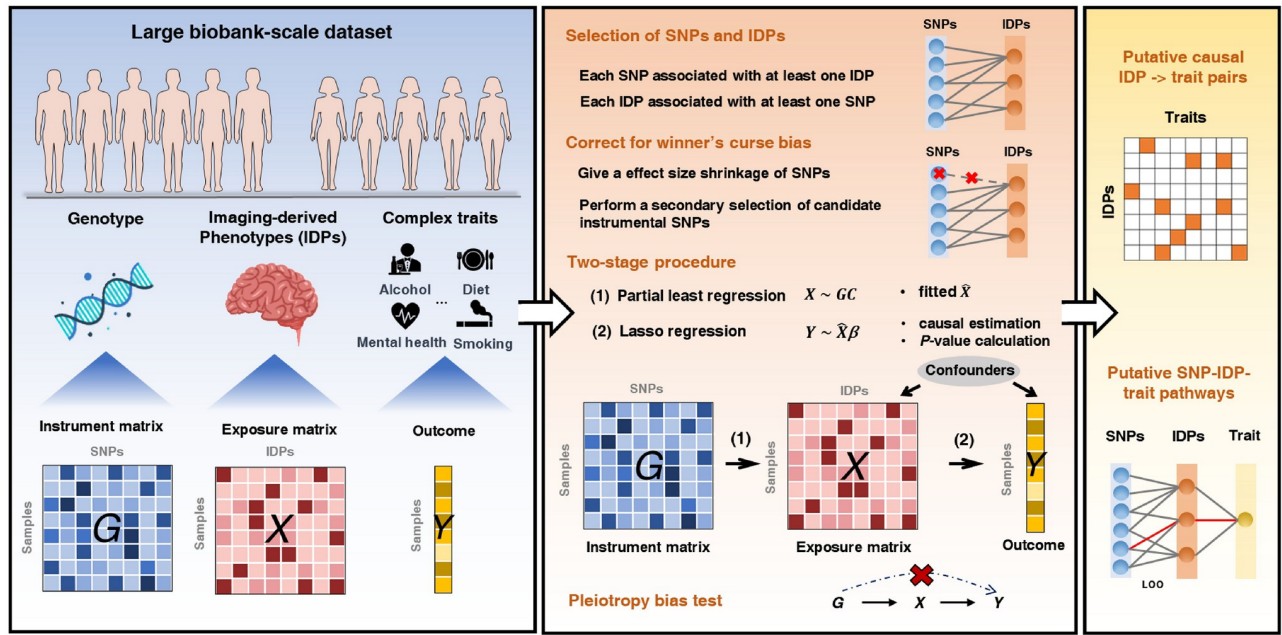

**Fig 1. Schematic showing the MR-PL method.** MR-PL is a one-sample multivariable MR framework that leverages data from a single large-scale dataset, including genotype, brain imaging data (or other correlated biomarkers, e.g., metabolites) and complex traits or diseases. A two-stage procedure is employed in MR-PL to test the causal association of exposures on the outcome, which considers the correlation among numerous exposures. To overcome the winner's curse (WC) bias problem, a WC-correction is performed when the same sample for instrument selection is adopted for MR analysis. MR-PL was employed for the identification of putative causal relationships of 36 white matter tracts on 180 complex traits in the UKBB. More details are presented in **Methods**.

microbiome). Given the prevalence and maturity of biobank-scale datasets for brain imaging data compared to other omics datasets, the focus was placed on IDPs that were adopted as the exposures in this study. Schematic of the proposed method is shown in Fig 1. In brief, a two-stage procedure is applied in MR-PL, where the exposure matrix $X$ is first regressed with instrumental matrix $G$ through partial least squares (PLS) regression [22], and the outcome is subsequently regressed on the predicted exposure matrix from the first stage by Lasso regression [23]. For a better ranking on the exposures in accordance with their strength of causality evidence, a regularized projection method termed de-sparsified Lasso is adopted to calculate the $P$-values of the exposures [24]. It is noteworthy that the proposed method can further make statistical correction of the winner's curse (WC) phenomenon in MR. This correction is determined by the $P$-value threshold employed for instrumental variable (IV) selection and the sample size, and it leads to stronger shrinkage for the GWAS association of IV-exposure pairs when the threshold is looser and sample size is smaller. This WC-correction procedure is computationally attractive for a number of IVs and a range of GWAS $P$-value thresholds, and only the $P$-values of SNPs from the GWAS summary statistics data are required in this step. In general, MR-PL enables a more realistic detection of the true causal effects from a number of correlated exposures in a single biobank-scale dataset compared with other approaches. More details of MR-PL are presented in **Methods**.

## Simulation results show better performance for MR-PL over other approaches

To evaluate the performance of MR-PL, the individual-level data of independent SNPs, exposures, and the outcome were simulated across a variety of scenarios, covering different

exposures heritability $hg^2$ (from low to high), different interference strengths of confounding factors on exposures $\alpha_{ux}$ (low-power and high-power), and different types of true causal effects $\beta_{xy}$ (discrete and continuous; weak, moderate, and strong) (see S1 Table for all parameter settings). Each parameter setting was repeated 100 times. The data generation mechanism in simulation is depicted in S1 Fig. Prior to MR analysis, a GWAS was performed for the respective exposure to obtain the marginal *P*-values between SNPs and exposures. SNPs with a marginal *P*-value $< 5 \times 10^{-8}$ linked to at least one exposure were selected as the initial instruments. MR-PL was compared with eight one-sample approaches including: (1) 2SLS-based methods: multivariable 2SLS (Multi-2SLS), univariable 2SLS (Uni-2SLS), polygenic risk score-based 2SLS (PRS-2SLS), and multivariable imaging wide association study (MV-IWAS) [25]; (2) a MR framework that jointly selects IVs and exposures: ImagingMR [20]; (3) three approaches that replace the second stage of MR-PL (Lasso) with other regularized regression methods: PLS-Ridge, PLS-Elasticnet and PLS-Lars. Notably, to perform a fair comparison with other approaches, the step of WC-correction in MR-PL was not included in this section. More details of the simulation analysis can be found in **Methods**.

First, we investigated how well the above-mentioned methods can estimate the causal effects, which was measured by the mean squared error (MSE) of the estimates. The MSE was defined as the mean squared difference between the true causal effect and estimated casual effect across all exposures, and was then averaged over 100 replications under each parameter setting. We found that MR-PL performed better than other methods in terms of MSE under almost all scenarios (Fig 2 and S2 Table). The next optimal approaches are PLS-Elasticnet, PLS-Ridge and PLS-Lars, indicating that the use of PLS regression in the first stage can better incorporate the correlation between exposures, potentially leading to improved causal effect estimation in the second stage. In certain circumstances when the causal effect $\beta_{xy}$ is drawn from set {-0.3,0,0.3}, the discrete and strong causal effect setting, Multi-2SLS outperformed other PLS + regularized regression approaches. This is presumably because the use of regularized regression leads to more estimate shrinkage when the true causal effect is stronger. Multi-2SLS can achieve lower MSE than Uni-2SLS, reflecting that incorporating multiple exposures into the model allows better accounting for variants with pleiotropic effects on exposures. MV-IWAS exhibited consistently enhanced MSE performance compared to Multi-2SLS and Uni-2SLS (Fig 2 and S2 Table), suggesting that leveraging only exposure-specific IVs for each exposure prediction in multivariable MR might further enhance causal estimation. PRS-2SLS performed the worst, with the MSE being thousands of times greater relative to other methods. This is expected as aggregating the allele scores of multiple instrumental SNPs into a single score will result in a significant loss of genetic information. All the above-described methods yielded lower MSE when the heritability $hg^2$ is higher and the interference strength of confounding factors $\alpha_{ux}$ is weaker. Given MSE is a combination of bias and variance, we also provided separate simulation results for bias and variance of causal estimation. We observed that MR-PL typically yielded lower values for both bias and variance compared to other approaches in nearly all settings, followed by PLS-Elasticnet (S2 Fig and S3 Table). These results indicate that the benefit of MR-PL for causal estimation lies in both bias and variance. In supplementary simulation with the presence of linkage disequilibrium (LD) among instrumental SNPs, the overall simulation results of MSE were similar, and the relative performance rank remained nearly unchanged (S4 Table), with MR-PL generally outperforming all other approaches. In supplementary simulation with the presence of pleiotropy, the comparison was narrowed among the most focused approaches of MR-PL, Multi-2SLS and Uni-2SLS, where MR-PL slightly outperformed other methods in terms of MSE at most parameter settings (S5 Table).

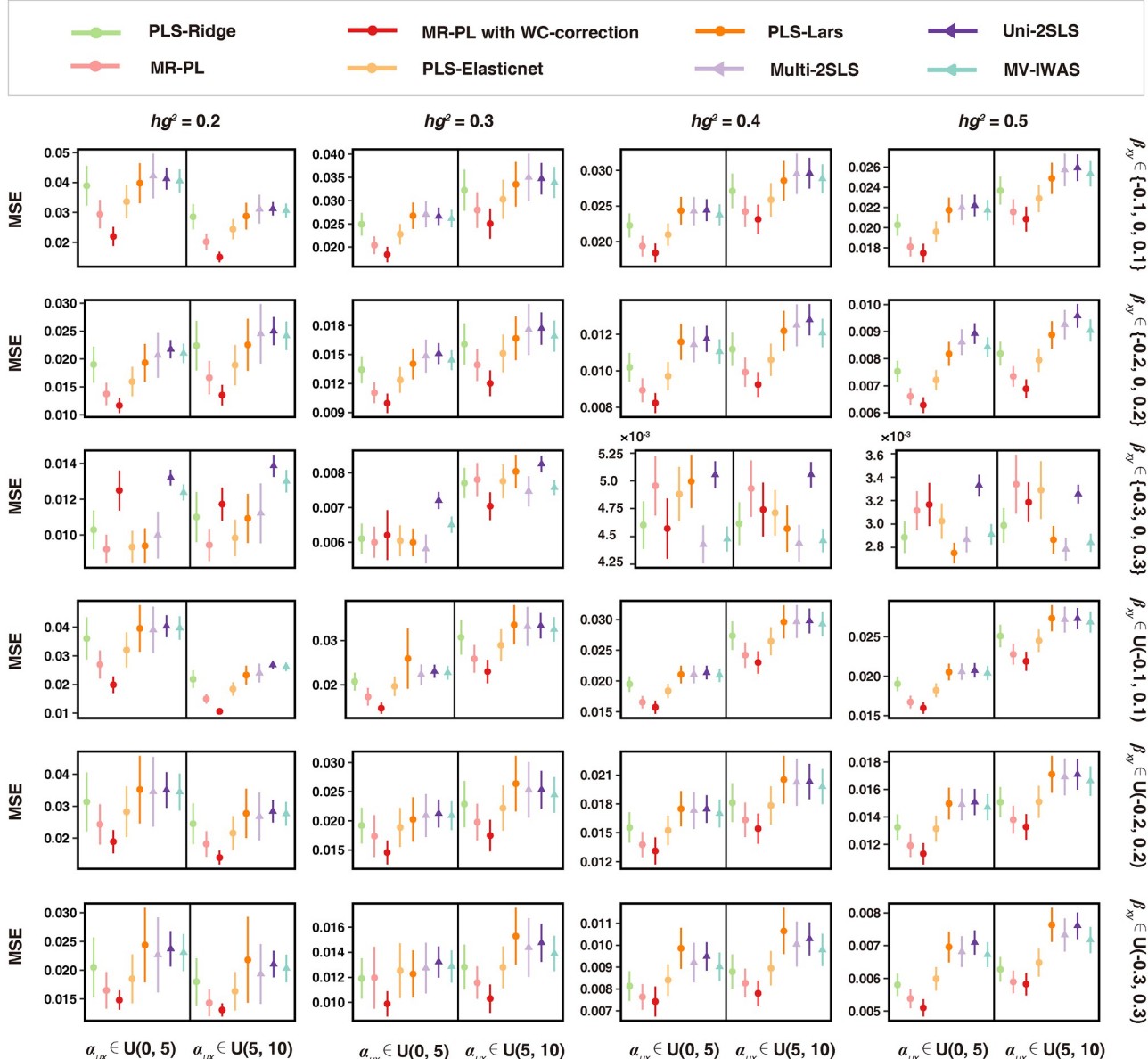

**Fig 2. Mean squared error (MSE) of MR-PL and other competing approaches.** The parameters were set to cover different exposures heritability $hg^2$, different interference strength of confounding factors $\alpha_{ux}$ and different types of causal effects $\beta_{xy}$ (S1 Table). Under each parameter setting, MSE was averaged over 100 replications. The error bar represents ten-fold variance of the MSE over 100 replications. For settings with too small variance, the error bar tends to degenerate to a point. Notably, the results from the methods of ImagingMR and PRS-2SLS were not included here because the MSE values from the two methods were extremely too large to show here. The full results can be found in S2 Table.

Next, type I error control of the approaches that can rank the importance of exposures in accordance with $P$-values (i.e., MR-PL, Multi-2SLS, Uni-2SLS, PRS-2SLS and MV-IWAS) were examined. The type I error rate was derived at 5% nominal level and averaged over 100 replications for each parameter setting. MR-PL stood out for the power to control type I error rate than other methods in almost all sceneries (Fig 3A). MR-PL achieved more false detections in scenarios when $\beta_{xy}$ was drawn from set {-0.1,0,0.1} (weak causal effects setting, S1 Table). This is expected as the Lasso regression tends to provide sparse solution, making it

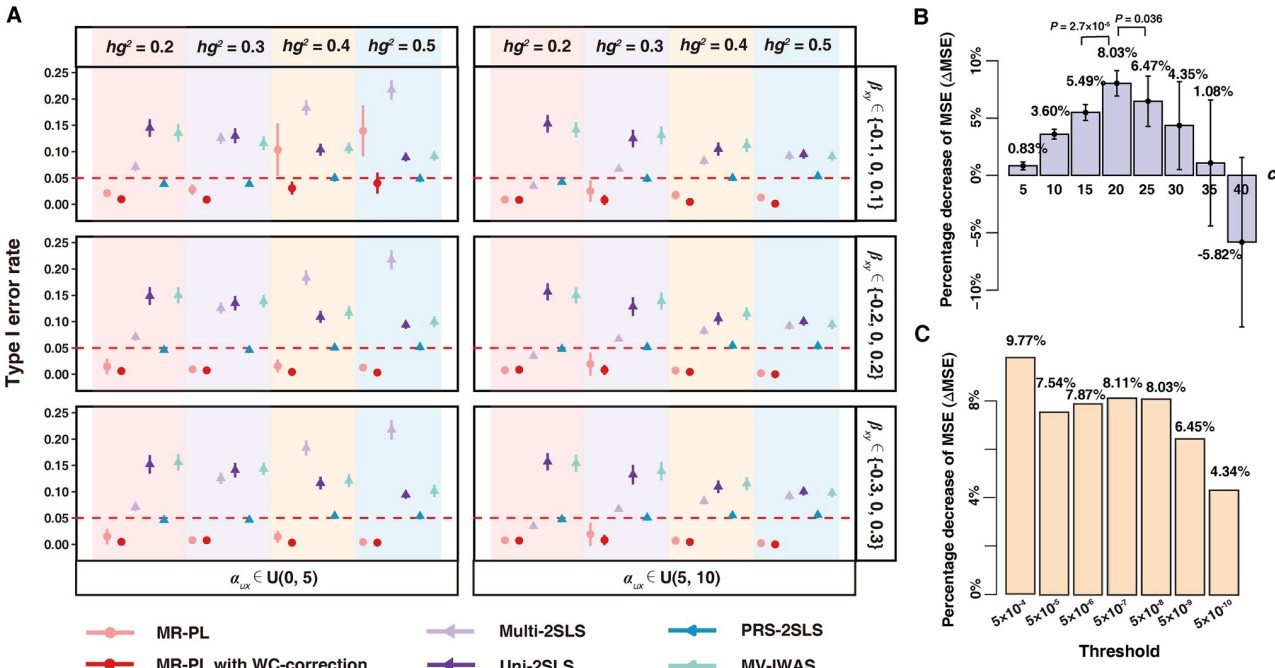

**Fig 3. Simulation results of MR-PL. (A)** A comparison of type I error rate across MR-PL and other competing approaches. Simulation settings were included if the causal effect was drawn from the discrete set. MR methods were included if individual *P*-value of causal effect estimate of the respective exposure can be calculated. A method failed to control type I error if its type I error rate exceeded the 0.05 nominal significance level (red dotted line). The error bar represents twice variance in type I error rate over 100 replications. Notably, the error bar tends to degenerate to a point for those settings with too small variance. **(B)** The ΔMSE index under different tuning parameter *c* ranging from 5 to 40 (step by 5). ΔMSE denotes the percentage decrease in MSE from MR analysis without WC-correction to MR analysis with WC-correction, averaged across all parameter settings. With the increase of ΔMSE, the greater the MSE was decreased after WC-correction. The Wilcoxon signed rank test was used to test the difference of the percentage decrease in MSE between *c* = 20 and its neighboring selections of 15 and 25 across all 48 parameter settings. **(C)** The percentage decrease of MSE (i.e., ΔMSE) under different GWAS *P*-value thresholds to select instrumental variants.

difficult to discern between causal effects that are to each other. Followed by MR-PL, PRS-2SLS exhibited overly stable type I error rate control of roughly 5% but at the expense of a large bias in causal effect estimation (S2 Table). Multi-2SLS, Uni-2SLS and MV-IWAS failed to control type I error rate, and exhibited inflated type I error rates across scenarios. In the presence of LD, MR-PL still controlled type I error rate better than other approaches, and an inflated type I error rate was identified for other approaches (S3 Fig). For the use of MR-PL, using independent variants as IVs was recommended for presenting an overly better power to estimate the causal effects and control type I error than in the presence of LD. In the presence of pleiotropy, all these approaches exhibited a considerably inflated type I error rate (larger than 0.2) (S4 Fig) when compared to simulations where no pleiotropy exists (Fig 3A). Despite this, the Sargan test used in our method successfully distinguished between scenarios with and without pleiotropy across all parameter settings (S5–S8 Figs). Therefore, we are able to further remove significant MR results that may be attributed to pleiotropy through combining the established pleiotropy-detecting capability of Sargan test with our method.

Power results (represented as 1-type II error rate at 5% nominal level) of approaches were provided in S9 Fig. We observed that all the MR approaches had similar power results, except PRS-2SLS which had low power under 0.5 across all settings. The highest power results can be seen for Multi-2SLS, but at a cost of inflated type I error rate and large MSE (Figs 2 and 3A). MR-PL displayed overall high-power results ranging from 0.975 to 0.998, highlighting its

ability to maintain substantial power while effectively controlling the type I error rate. These results suggest that our method still remains the sensitivity to detect true causal relationships while controlling for type I error rate and minimizing MSE best.

## Winner's curse correction further improves the performance of MR-PL

The WC phenomenon refers to the tendency of associations to be overestimated in the discovery population due to chance noise [7]. WC can be identified when the identical sample for IV selection is employed for MR analysis, thus triggering biased causal effect estimation [7]. Using another independent sample to select IVs can avoid WC bias, whereas some limitations remain, especially in multivariable MR. We introduced a procedure for WC-correction in MR-PL, correcting for the GWAS association of SNP-exposure pairs in accordance with the *P*-value threshold used for IV selection and sample size. The overly correction strength is adjusted through the tuning parameter *c*. The index ΔMSE was used for the choice of optimal *c*, where ΔMSE denotes the percentage decrease in MSE from MR analysis without WC-correction to MR analysis with WC-correction. In simulation, *c* varied from 5 to 40 (step by 5), and the value with the highest average ΔMSE across all parameter settings at *P*-value threshold $5\times10^{-8}$ was selected.

When *c* was set to 20, MR-PL exhibited the most significant percentage decrease in MSE of 8.03% (Fig 3B). As depicted in Fig 2 and S2 Table, MR-PL with WC-correction (*c* = 20, not specified below) can further lower MSE across almost all scenarios when compared with the absence of WC-correction. Such reduction was also applied to both bias and variance (S2 Fig and S3 Table). As shown in Fig 3A, the statistical capacity of MR-PL with WC-correction for controlling type I error rate has also been enhanced, particularly for the scenarios with an inflated type I error rate when no WC-correction was applied. Implementation of WC-correction increased the power of MR-PL, except for scenarios with low heritability ($hg^2$ = 0.2) where the power slightly reduced (S9 Fig). Similar results were reported for neighboring selections of *c* at 15 and 25 (S10–S15 Figs), indicating that performance improvement achieved via WC-correction procedure is robust across a gird of neighboring *c* values. Furthermore, the simulation was expanded to cover different *P*-value thresholds for instrument selection including $5\times10^{-4}$, $5\times10^{-5}$, $5\times10^{-6}$, $5\times10^{-7}$, $5\times10^{-9}$ and $5\times10^{-10}$, since different thresholds have been often used for reasons such as variants achieving genome-wide significance level lack sufficient power to predict the exposure [26]. Under different *P*-value thresholds, MR-PL with WC-correction achieved lower MSE and type I error rate compared with the absence of WC-correction, except for some circumstances where $\beta_{xy}$ was drawn from set {-0.3,0,0.3} (S16 and S17 Figs). As depicted in Fig 3C, WC-correction tended to reduce MSE more when the threshold was more relaxed (MSE decreased by 4.34% ~ 9.77% for thresholds ranging from $5\times10^{-4}$ to $5\times10^{-10}$), suggesting that rigorous thresholds are conducive to alleviating the WC phenomenon. The above conclusion that WC-correction can improve the performance of MR-PL in reducing MSE and type I error rate across various *P*-value thresholds for IV selection still held for neighboring selections of *c* at 15 and 25 (S18–S21 Figs), as well as in supplementary simulation with the presence of LD (S4 Table, S3, S22 and S23 Figs).

Additionally, we examined the Sanderson-Windmeijer conditional F-statistics ($F_{sw}$) values in our simulations [18,27] (see S1 Text for more details). It shown that even in our ideal simulation with sufficient exposure-specific instruments, the $F_{sw}$ values can hardly reach the rule-of-thumb value of 10 (with a mean 4.37 before WC-correction, and 5.63 after WC-correction at *c* = 20, S24A Fig and S6 Table). To investigate the simulation performance under different levels of conditional F-statistics, we segmented the corresponding $F_{sw}$ values into distinct bins for all the 43,200 simulations (100 replications × 48 settings × 9 *c* values varying from 0 to 40).

It was observed that the $F_{sw}$ values were strongly negatively correlated with the number of instruments and exposures (S24B–S24E Fig). Moreover, simulations falling within the bin with $F_{sw}$ values excessed 10 gave the poorest performance in MSE and type I error control in our method (S24F and S24G Fig). These indicate the potential limitation of the conditional F-statistic in application of high-dimension and high-correlation situations.

## Real data application of MR-PL to investigate the causal effects of IDPs on complex traits

To verify the utility of the proposed method in the real data, MR-PL was applied to individual-level data in the UKBB to predict causal effects between 36 types of white matter microstructure (measured by the fractional anisotropy, S7 Table) and 180 complex traits, with sample size ranging from 2,087 to 32,666 (S8 Table and S25 Fig). It has been shown that genetic factors significantly affected the variation of white matter microstructure [28,29]. Almost half of the SNP-based heritability and hundreds of significant genetic loci were found by two GWASs on white matter microstructures [28,29]. Furthermore, the shared genetic associations between white matter structures and a wide spectrum of complex traits were revealed [28–30]. Thus, the causal relationships between white matter tracts and diverse complex traits were investigated using the proposed method.

The 180 complex traits were divided into ten categories for better interpretability, including biochemical, physical, cognitive, health related, eating, lifestyle, alcohol intake, smoking, mental health related and socioeconomic categories. Independent variants ($r^2 < 0.1$) reaching genome-wide-significance ($P < 5 \times 10^{-8}$) were selected as the initial instrument based on a published GWAS on brain white matter microstructure [28], resulting in 368 independent variants for 36 associated white matter tracts (i.e., white matter tracts whose FA values were associated with at least one genetic variant) (S9 Table). After the WC-correction with parameter $c = 20$, totaling 69 variants for 34 associated white matter tracts were retained for MR analysis. The F-statistic and conditional F-statistic for the respective IDP exposure were provided in S10 and S11 Tables, respectively.

After discarding four IDP-trait pairs where horizontal pleiotropy exists ($P < 0.05$, Sargan test), we identified 233 IDP-trait pairs, for which 32 white matter tracts have a nonzero causal effect on 130 different traits (Fig 4A and S12 Table). Among these, we eventually retained 26 IDP-trait pairs with $P$-value less than 0.05, and seven of them with false discovery rate (FDR) less than 0.1 (Table 1 and Fig 4A), including causal associations between IDPs with five eating behaviors, four smoking behaviors, four biochemical measures, two health related traits, four physical measures, two cognitive traits, two socioeconomic measures and two lifestyle traits. We found that most (18 out of 26) of these predicted causal associations could be supported by previous literature (S13 Table), suggesting the high reliability of the results generated by MR-PL.

## Putative causal white matter tracts implicated in smoking, blood vascular function-related traits, and eating behaviors

We found that the traits categorized in smoking could be putatively affected by three IDPs, i.e., the FA values of the right superior fronto-occipital fasciculus (SFO), right retrolenticular part of internal capsule (RLIC), and left posterior thalamic radiation (PTR) (Fig 4B and Table 1). The right SFO exhibited dominate causal association with smoking traits, where lower FA value of right SFO could be causally associated with more serious smoking behavior (pack years of smoking: $\beta = -0.005$, $P = 6.85 \times 10^{-5}$; pack years adult smoking as proportion of life span exposed to smoking: $\beta = -0.032$, $P = 5.04 \times 10^{-4}$). The SFO is a fiber bundle linking the

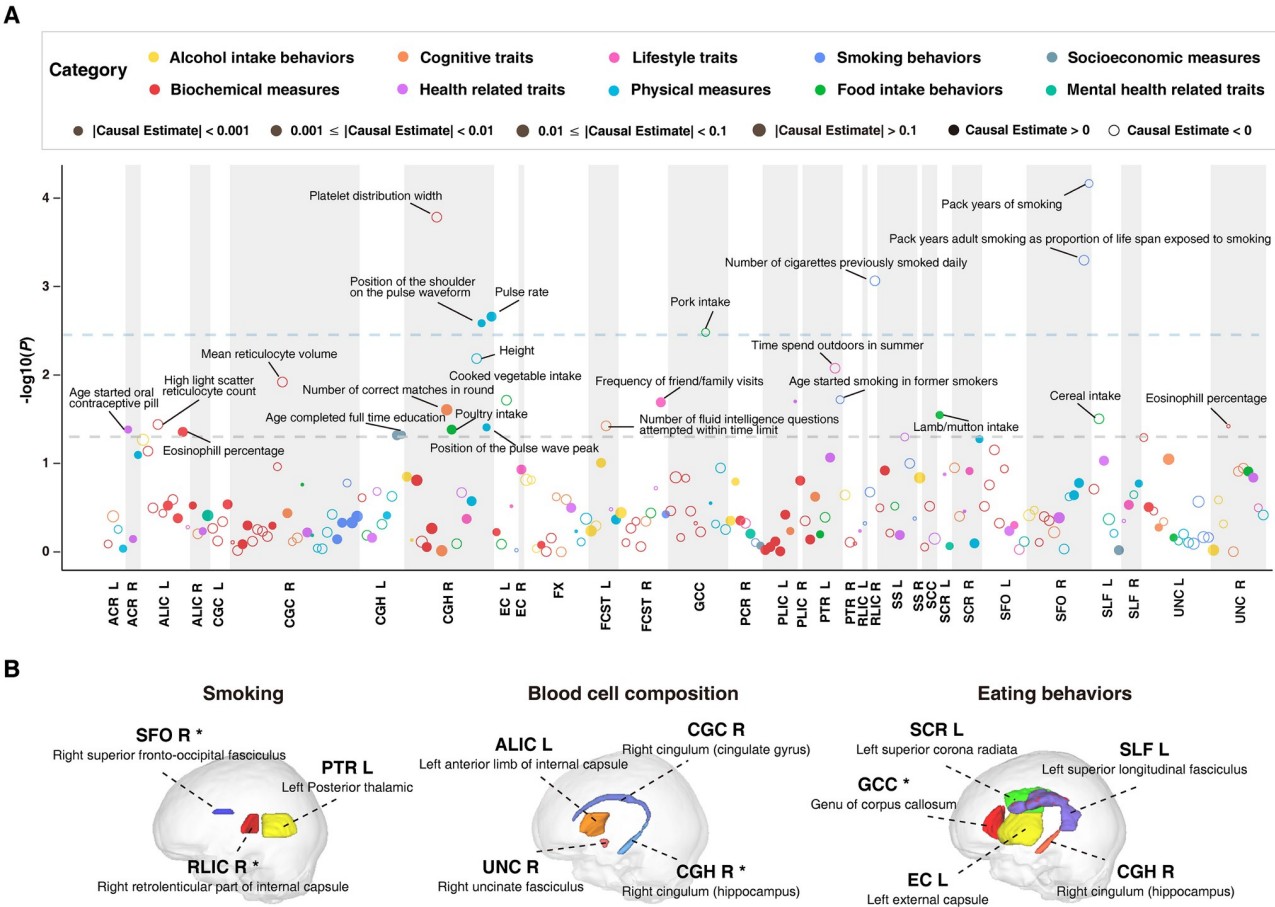

**Fig 4. MR results for pairs of IDPs and complex traits. (A)** Manhattan plot showing the significant levels of causal associations between 34 IDPs and 130 complex traits with a nonzero causal effect estimate. The point above the grey dotted line denotes a complex trait with *P*-value < 0.05. The point above the blue dotted line denotes a complex trait with FDR < 0.1. **(B)** White matter tracts associated with different traits at *P*-value < 0.05, including right superior fronto-occipital fasciculus, left posterior thalamic radiation and right retrolenticular part of internal capsule associated with traits categorized in smoking (left); left anterior limb of internal capsule, right cingulum cingulate gyrus, right uncinate fasciculus and right cingulum hippocampus associated with traits categorized in blood cell composition (middle); genu of corpus callosum, left external capsule, left superior corona radiata, left superior longitudinal fasciculus and right cingulum hippocampus associated with traits categorized in eating behaviors (right). The asterisks denote MR results with FDR < 0.1.

superior frontal gyrus and superior parietal lobe [31], and may play key roles in visual processing function [32,33]. The abnormalities of the white matter microstructure of SFO have been frequently reported in smokers [34,35]. Our results indicate that the individuals with poor white matter integrity in SFO are more sensitive to external visual stimulation of cigarettes, thus tend to have longer smoking duration. The RLIC largely contains fibers of the optic radiation that links the lateral geniculate nucleus with the calcarine fissure [36], and the PTR is a bundle of projection fibers that links the caudal parts of thalamus with the parietal and occipital lobe [37]. Both of the fiber tracts have been reported for their implications in smoking [38,39].

Our findings offer novel insights into the underlying causal associations between IDPs and blood vascular function-related traits. The mean FA value of right cingulum (hippocampus part) (CGH) showed to have putative causal effects on arterial stiffness-related traits in category of physical measures (pulse rate: $\beta$ = 0.089, *P* = 2.19×10$^{-3}$; position of the shoulder on the

**Table 1. Summary of MR-PL results for pairs of IDPs and complex traits with a non-zero causal effect estimate at *P*-value < 0.05 and no horizontal pleiotropy (*P*-value > 0.05, Sargan test).**

| Category of traits | Trait | White matter IDP | Causal Estimate | P |
|---|---|---|---|---|
| Smoking behaviors | Pack years of smoking | superior fronto-occipital fasciculus R | -0.005 | 6.85E-05 |
| | Pack years adult smoking as proportion of life span exposed to smoking | superior fronto-occipital fasciculus R | -0.032 | 5.04E-04 |
| | Number of cigarettes previously smoked daily | retrolenticular part of internal capsule R | -0.100 | 8.62E-04 |
| | Age started smoking in former smokers | posterior thalamic radiation L | -0.002 | 1.90E-02 |
| Biochemical measures—blood cell composition | Platelet distribution width | cingulum hippocampus R | -0.066 | 1.64E-04 |
| | Mean reticulocyte volume | cingulum cingulate gyrus R | -0.018 | 1.20E-02 |
| | High light scatter reticulocyte count | anterior limb of internal capsule L | -0.073 | 3.63E-02 |
| | Eosinophill percentage | anterior limb of internal capsule L | 0.036 | 4.40E-02 |
| | Eosinophill percentage | uncinate fasciculus R | 0.0003 | 3.79E-02 |
| Physical measures—Arterial stiffness | Pulse rate | cingulum hippocampus R | 0.089 | 2.19E-03 |
| | Position of the shoulder on the pulse waveform | cingulum hippocampus R | 0.009 | 2.60E-03 |
| | Position of the pulse wave peak | cingulum hippocampus R | 0.006 | 3.91E-02 |
| Physical measures | Height | cingulum hippocampus R | -0.034 | 6.52E-03 |
| Cognitive traits | Number of correct matches in round | cingulum hippocampus R | 0.112 | 2.47E-02 |
| | Number of fluid intelligence questions attempted within time limit | fornix cres+stria terminalis L | -0.043 | 3.76E-02 |
| Eating behaviors | Pork intake | genu of corpus callosum | -0.003 | 3.30E-03 |
| | Cooked vegetable intake | external capsule L | -0.015 | 1.93E-02 |
| | Lamb/mutton intake | superior corona radiata L | 0.007 | 2.84E-02 |
| | Cereal intake | superior longitudinal fasciculus L | -0.066 | 3.13E-02 |
| | Poultry intake | cingulum hippocampus R | 0.043 | 4.14E-02 |
| Health related traits | Age at last live birth | posterior limb of internal capsule L | 0.0002 | 1.99E-02 |
| | Age started oral contraceptive pill | anterior corona radiata R | 0.006 | 4.13E-02 |
| Lifestyle traits | Time spend outdoors in summer | posterior thalamic radiation L | -0.022 | 8.39E-03 |
| | Frequency of friend/family visits | fornix cres+stria terminalis R | 0.035 | 2.04E-02 |
| Socioeconomic measures | Age completed full time education | cingulum hippocampus L | 0.004 | 4.74E-02 |
| | Average total household income before tax | cingulum hippocampus L | 0.011 | 4.80E-02 |

pulse waveform: $\beta = 0.009$, $P = 2.6\times10^{-3}$; position of the pulse wave peak: $\beta = 0.006$, $P = 3.91\times10^{-2}$). Arterial stiffness is a leading marker for cardiovascular events and neurovascular dysfunction, which is measured through index of pulse like pulse wave velocity or pulse pressure [40,41]. The CGH is a cortico-limbic tract connecting the cingulate gyrus and parahippocampal portion of entorhinal cortex [33]. Our results are in line with the previous research that identifying an association between abnormalities in parahippocampal cingulum and arterial stiffness [42]. Moreover, several IDPs were found to have putative causal effect on blood cell composition parameters, including the mean FA value of CGC on platelet distribution width ($\beta = 0.066$, $P = 1.64\times10^{-4}$), and the mean FA value of cingulum cingulate gyrus on mean reticulocyte volume ($\beta = 0.018$, $P = 1.2\times10^{-2}$). Two IDPs, the mean FA of the right uncinate fasciculus and anterior limb of the internal capsule, were found to be nominally associated with eosinophil percentage. It was reported that white matter lesions can trigger blood–brain barrier alterations [43], a plausible mechanism linking white matter damage and inflammation [44]. Such alterations will facilitate the migration of leukocytes, including eosinophils, within

the bloodstream [45]. We expect that microstructure changes in these two tracts could contribute to blood–brain barrier alterations, thus potentially leading to eosinophils changes in blood. By adding new confounding factors as covariates which are potentially linked with blood cell composition and physical measures, including LDL cholesterol, HDL cholesterol, triglycerides, alcohol intake frequency and pack years of smoking, the above causal association pattern still remains (S14 Table). Despite rare literature illustrating the correlation between individual white matter tracts and blood cell count, there is emerging evidence for the correlation between global measures of brain white matter and blood cell count [46,47]. Since brain structural imaging markers (e.g., white matter integrity) offers a clue for early detection of cardio-cerebrovascular diseases (e.g., strokes) [48,49], and hemogram parameters (e.g., platelet distribution width) serve as vital prognostic markers for cardio-cerebrovascular diseases [50,51], the findings of this study may provide useful references for in-depth research on the diagnosis propagation path in this aspect.

In addition, the results suggest that several IDPs were predicted to show putatively causal effects on the traits categorized in eating behaviors, comprising the FA values of the left external capsule (EC), left superior longitudinal fasciculus (SLF), genu of corpus callosum (GCC), and left superior corona radiata (SCR). Appetitive behaviors are regulated by brain reward processing circuits that integrates hunger signals, taste stimulation, and cognitive-emotional processes [52]. The EC, SLF, GCC, and SCR are all taste-reward-related white matter tracts. To be specific, the corpus callosum, a large bundle of fibers coursing through the left and right hemispheres, has been reported to be associated with interhemispheric transferring taste signal [53]. The EC refers to a bundle of fiber tracts that pass through the uncinate fasciculus and inferior fronto-occipital fasciculus [54], and the SLF is a bundle of long association fibers that course alongside the superior edge of the insula, connecting the frontal, temporal, parietal, and occipital lobes [55]. White matter tracts in the above-mentioned pathways have been reported to be associated with brain reward activation [56]. Lesions in corona radiata were indicated in altered taste perception [57]. Moreover, abnormal microstructural integrity of the corpus callosum, EC, SCR and SLF has been frequently found in women with eating disorders (e.g., anorexia nervosa) [58–60]. These results can support previous hypothesis that white matter fibers functioning in reward processing could be biological markers that drives food intake behaviors, while providing useful references for designing brain stimulation strategies to modulate appetitive behaviors in treatment of eating disorders [61,62].

Next, the MR-PL analysis was repeated for correlated variants ($r^2 < 0.6$) as instrument (S15 Table), and 223 IDP-trait pairs having nonzero causal effect estimates were identified (S16 Table). To be specific, 153 IDP-trait pairs (68.6%) exerted a nonzero causal effect $\beta$ estimation by both types of instrument selection. Their pattern of $\beta$-maps (a vector that containing all non-zero causal estimates of IDPs on traits) was highly consistent ($r = 0.91$, $P < 5\times10^{-12}$, Spearman correlation) between the results from the two instrument selection strategies, and the associations between IDPs and smoking, blood cell count and eating behaviors were still identified at $\beta \neq 0$ and $P < 0.05$, suggesting that the proposed method can be robust to different correlating structure of instrumental variants. Analogously, significant positive correlation of $\beta$-maps were observed between the above main MR-PL results with WC-correction at $c = 20$ and MR-PL results with WC-correction at neighboring selections of $c = 15/25$ ($r = 0.831/0.851$, both with $P < 5\times10^{-12}$, Spearman correlation, S17 and S18 Tables), as well as MR-PL results without WC-correction ($r = 0.792$, $P < 5\times10^{-12}$, Spearman correlation, S19 Table). Results of Uni-2SLS and Multi-2SLS are also provided in S20 Table. More strong causal effects were obtained in Uni-2SLS than Multi-2SLS, in line with the simulation results that Uni-2SLS generally gave more inflated type I error rate than Multi-2SLS (Fig 3). A consistent trend in causal estimations was observed across MR-PL, Uni-2SLS and Multi-2SLS ($r = 0.33$ between

MR-PL and Multi-2SLS IDP-trait $\beta$-maps, $r = 0.53$ between MR-PL and Uni-2SLS IDP-trait $\beta$-maps, $r = 0.27$ between Multi-2SLS and Uni-2SLS IDP-trait $\beta$-maps, all with $P < 5\times10^{-5}$, Spearman correlation). Furthermore, MR-PL can reveal potential causal relationships that were not found by Uni-2SLS or Multi-2SLS. For example, Uni-2SLS failed to reveal the eight potential causal relationships between cingulum hippocampus and different traits shown in Table 1. This is possibly because only 11 SNPs were used as IVs to estimate the causal effect of cingulum hippocampus on traits in Uni-2SLS, and the limited number of SNPs may not be adequate to capture its variation given the reported high heritability of cingulum hippocampus (46%~57% [28,63]). Multi-2SLS failed to detect the well-known association between SFO and smoking ($P$-value ranging from 0.43 to 0.97 for all SFO-smoking pairs, S20 Table). However, other highly correlated tracts such as fornix ($r = 0.48/0.55$ between fornix and right/left SFO) exhibited significant causal relationship with traits categorized in smoking, which is less evidenced for their association by prior literature. This is possibly because Multi-2SLS is always prone to weak instrument bias under high-dimensional and high correlation conditions since it doesn't consider the influence of other correlated exposures in the first stage of exposure prediction.

## Putative causal SNP-IDP-trait pathways implicate regulatory mechanisms in complex traits

Unlike canonical two-sample MR, in which population characteristics differ significantly between samples and thus affect results interpretation, one-sample MR contributes to the exploration of biological pathways in a single homogenous dataset. Integrating SNP-IDP pairs with IDP-trait pairs into a SNP-IDP-trait pathway from individual-level data can provide insights into the regulatory mechanisms underlying these complex traits. We conducted a leave-one-out cross-validation (CV) strategy in PLS regression at the first stage of MR-PL, and identified 435 significant SNP-IDP pairs between 53 SNPs and 16 IDPs (at a Bonferroni-corrected $P$-value threshold of $2.13\times10^{-5} = 0.05/34/69 = 0.05$ / number of IDPs retained after the WC-correction / number of variants retained after the WC-correction). After combined with the above 26 IDP-trait pairs with a non-zero causal effect at $P < 0.05$, we constructed 224 SNP-IDP-trait propagation paths (S21 Table). The respective instrumental SNP was mapped to its nearby genes within 10kb, and thorough searches of literature were conducted in the NHGRI-EBI GWAS Catalog (version 2022-09-30) [64] to gain insights into the functional roles of the above-mentioned genes.

Here, a predicted SNP-IDP-trait pathway was exemplified: rs76122535—right SFO—Smoking (Pack years of smoking). The SNP rs76122535 lies in an intronic region within the protein coding gene *ICA1L*, which encodes a member of the interacting BAR-domain family of proteins [65], and has been implied in impaired excitatory synaptic signaling [66]. Existing studies suggested that the *ICA1L* gene is associated with the microstructure of SFO [29] and smoking status [67]. As indicated by our results, rs76122535 (*ICA1L*) may affect the white matter microstructure of SFO, making individuals more sensitive to external visual stimulation of cigarettes, thus tending to have longer smoking time.

Another interesting example of the pathway is rs17205972—right CGC—platelet distribution width. The SNP rs17205972 lies in an intronic region within the protein coding gene *VCAN*, which encodes versican, a large chondroitin sulfate proteoglycan. The *VCAN* gene takes on critical significance in tissue morphogenesis and maintenance, and has been well known for its association with brain white matter including the fiber tract of CGC [30]. Besides, *VCAN* is correlated with hemogram parameters (e.g., the monocyte count [68] and platelet-derived growth factor [69]). As indicated by the identified coherent biological

pathway, rs17205972 (*VCAN*) may contribute to the causal effect of right GCG on the platelet distribution width. In S22 Table, we present a summary of the pathways with SNP-mapped genes and GWAS Catalog traits, which contains detailed information and can be further studied to decipher the causal regulatory mechanisms from genetic variants to complex traits.

## Discussion

In this study, MR-PL, a one-sample multivariable MR method, is proposed to infer the causal relationships of multiple correlated candidate exposures on the trait. MR-PL also corrects for the winner's curse bias caused by the instrument selection process. Compared with other methods, MR-PL yielded lower estimation bias and type I error rate in various scenarios. The estimation bias can be further reduced to a large proportion when applying the step of WC-correction. We also proved its efficacy on the UKBB dataset across 36 white matter tracts and 180 health-associated traits, and explored many meaningful results which help to elucidate the underlying neural mechanisms regulating complex traits. In this study, we primarily focused on white matter microstructure as exposures in the real data applications. It's noted that exposures can still be modified as other imaging modalities given the emerging causal relationships reported between different imaging modalities (e.g., brain volumes) and complex traits [3,25]. Furthermore, it's natural to apply MR-PL to other high-dimensional omics data such as metabolomes, proteomes and microbiomes.

In simulation, MR-PL consistently outperformed other competing approaches, no matter using independent SNPs or correlated SNPs with real LD structure as instruments, confirming its robustness for different LD structures. In the real data application, MR-PL also demonstrated convergent results for independent and correlated SNPs as instruments. The possible reason is that PLS regression is adopted to fit the model at the first stage of the two-stage inference procedure, accounting for the (residual) correlation of both SNPs and exposures as well as their covariance. Accordingly, the potential of genetic information can be fully exploited to explain exposures, leading to a gain of statistical capacity in the subsequent stage. However, the question of whether to select independent SNPs or correlated SNPs as instruments remains a pending issue in MR. Numerous MR methods have been developed for independent instrument setting (e.g., GSMR [70], CAUSE [71], and TWMR [12]) and correlated instrument setting (e.g., PMR-Egger [72], MRAID [73]). On the one hand, complex traits can be affected by multiple SNPs which are in potential LD with each other, such that incorporating correlated SNPs may be conducive to capturing more exposure variance. On the other hand, incorporating larger number of SNPs can elevate the risk of bias in MR estimations caused by weak instrument bias or pleiotropy, since some irrelevant SNPs are used as the predictors as well. In this study, the more conservative choice of independent SNPs as instruments was recommended, which also gained better control of type I error rate than correlated instrumental SNPs in simulations (Fig 3 and S3 Fig).

The issue of weak instruments is often raised in multivariable MR, which is arguably complex and challenging in high-dimension and high-correlation scenarios. According to our results, the conditional F-statistic $F_{sw}$ can hardly reach the rule-of-thumb value of 10 neither in our ideal simulations nor real data applications (S24A Fig, S6 and S11 Tables), and is highly sensitive to the number of instruments and exposures (S24B–S24E Fig). Forcing to make it larger than 10 by reducing the number of instruments or exposures could result in poor causal estimation results (S24E and S24F Fig). Furthermore, the current conditional F-test used to evaluate weak instrument strength is basically built within the framework of 2SLS, where exposures are fitted in a totally different manner with our method in the first stage (2SLS: separately and parallelly; MR-PL: iteratively and considering all exposures as a whole). These indicate the

potential limitation of the conditional F-statistic in evaluating instruments strength under one-sample, high-dimension and high-correlation setting, especially in our method. Currently, few studies have discussed this issue in cases of extreme upscaling the number of exposures and instruments, and most of existing studies just constrained to the analysis on a limited number of exposures. Incorporating dimensionality reduction techniques in MR or using only exposure-specific IVs as predictors might be one of the smart ways to solve this problem [25,74,75]. We believe that more robust and data-driven statistical tools to evaluate the instrument strength and post-MR refining methodologies would be developed in cases of extreme upscaling the numbers of exposures and instruments in the future.

The WC-correction procedure presented here only requires the GWAS *P*-values of SNPs and has shown generally improved MR results across diverse scenarios including different heritability, confounding factors strength, patterns of causal effects, correlating structure of instrumental SNPs and *P*-value thresholds for IV selection, as well as robust results in both simulation and real data application. Therefore, it can be fast and conveniently applied without adding any additional computational complexity. Theoretically, the WC-correction procedure won't change the overall tendency of MR results. For example, in simulations, both MR-PL with and without WC-correction gave low MSE under 0.0034 at parameter setting $hg^2$ = 0.5, $\beta_{xy} \in$ {-0.3,0,0.3}, $\alpha_{ux} \in$ U(0,5); and both gave large MSE excess 0.023 at parameter setting $hg^2$ = 0.3, $\beta_{xy} \in$ {-0.1,0,0.1}, $\alpha_{ux} \in$ U(5,10) (Fig 2). In the real data application, high causal estimations correlation between MR-PL results with and without WC-correction were observed. Here, we recommend the following selection of *c* in the real data application:

Step 1. Determine whether the WC-correction is required; if the IV selection is based on a different sample from that used for MR analysis, then the correction is not required;

Step 2. Choose the value for the parameter *c*. Here, we strongly recommend to select *c* as 20 based on our results;

Step 3: Examine the robustness of MR results at neighboring values of the selected *c* (e.g., 15 or 25).

The WC-correction procedure can be considered as a secondary selection of candidate instruments (and exposures). It can further refine the selection boundary, significantly reducing the number of falsely selected IVs (S26 Fig). This process tries to achieve a delicate balance between reducing the inclusion of non-causal SNPs while retaining sufficient genetic information for analysis. Consequently, this adjustment would mitigate potential bias in exposure prediction and reduce causal estimate inflation thereafter. It should be noted that our simple WC-correction is not optimal and is dependent on multiple factors, including the genetic architecture of exposures, correlation pattern among exposures, linkage disequilibrium pattern of instrumental SNPs, and the ancestry of population. In the future, more accurate WC-correction methods tailored to MR with high-dimension and high-correlation situations would be developed. With rapid accumulation of the biobank-scale datasets and GWAS summarized data, the WC bias issue can be entirely avoided by selecting IVs based on a different dataset [7].

There are some other issues need to be discussed. First, like many other MR methods, the causal relationship identified by MR-PL cannot be taken as an exact causality, where caution should be exercised for interpretation. Second, our method cannot deal with the pleiotropy itself, and it relies on Sargan test to identify scenarios with the presence of global horizontal pleiotropy. A more efficient way is to incorporate the pleiotropy term in the model like MV-IWAS-Egger and PMR-Egger [25,72]. In the future, we will take into account the pleiotropy by directly incorporating SNP effects into the second stage of our model. Third, MR-PL cannot

detect and remove individual outlier genetic variants. Unlike two-sample MR, where numerous methods have been developed for detection and removal of individual variants (e.g., MR-PRESSO outlier [76] and HEIDI-outlier test [70]), there has been rare research in one-sample (multivariable) MR field. In-depth research can design more reasonable procedures in outlier variants detections in one-sample multivariable MR field. Finally, in addition to the number of exposures and instruments, the magnitude of the parameter chosen in the simulation (e.g., $hg^2$), could be one of the strong predictors that guiding the instrumental strength. More detailed studies with a wider range of parameters can look into the factors affecting instrument strength.

To summarize, we developed MR-PL, a novel MR analysis framework, which can be used in causal effect estimation from multiple exposures in a single biobank-based dataset. Here, we exemplified the ability of MR-PL in prediction of causal effects of white matter tracts on the complex traits from the UKBB. We hope that the causality evidence predicted by our method can be useful for deciphering the mechanisms underlying complex traits and diseases.

## Methods

### One-sample multivariable Mendelian Randomization and the causal model

The aim of this study was to estimate and test the causal effects of a set of exposures on the outcome in the context of one-sample multivariable MR setting, where the genetic variants, exposures and the outcome are derived from the same population-based cohort. Following the multivariable MR assumption [11,19], a genetic variant serves as a valid instrumental variable (IV) when satisfying the following (A1)–(A3) criteria, and an exposure can be included in the analysis when satisfying the following (A4)–(A5) criteria:

(A1) The variant is associated with at least one of the exposures;

(A2) The variant is independent of any confounding factor for any of the exposure-outcome association;

(A3) The variant cannot be associated with the outcome conditional on exposures and confounding factors;

(A4) The exposure is associated with at least one of the variants;

(A5) The exposure cannot be explained by the linear combination of the other exposures.

For exposures wherein exists highly structural, functional, and genetic correlation (e.g., IDPs), variants are likely to be associated with the outcome through multiple correlated exposures, leading to horizontal pleiotropy in the MR analysis. In conventional univariable MR, only one exposure is included each time, thus triggering biased causal effect estimate and increased false positive rate. In contrast, multivariable MR allows for measured pleiotropy through any one of the exposures, which can better control for pleiotropy bias [13]. Thus, compared with univariable MR, it's more reasonable to use multivariable MR for those correlated phenotypes. Moreover, the use of individual-level data allows us to better account for the correlation among exposures in a homogenous data.

We denote $N$ as the sample size, $M$ as the total number of SNPs selected as IVs, $K$ as the total number of exposure variables. We also denote $\boldsymbol{G} = \{\boldsymbol{G}_1, \boldsymbol{G}_2,\ldots, \boldsymbol{G}_M\} = \{g_{ij}\}_{N\times M}$ as $N$ by $M$ genotype matrix, where $\boldsymbol{G}_m$ ($1\leq m\leq M$) denotes the $N$-vector observations of SNP $g_m$, $\boldsymbol{X} = \{\boldsymbol{X}_1, \boldsymbol{X}_2,\ldots, \boldsymbol{X}_K\} = \{x_{ik}\}$ as $N$ by $K$ exposure matrix, where $\boldsymbol{X}_k$ ($1\leq k\leq K$) denotes the $N$-vector observations of exposure variable $x_k$, $\boldsymbol{Y}$ denotes the $N$-vector observations of the outcome variable $y$. We model the association among the genotypes, exposures and outcome through the following

two-stage estimation procedure.

$$X = GC + E \tag{1}$$

$$Y = \hat{X}\beta + \varepsilon \tag{2}$$

where $C$ denotes the coefficient matrix of $G$ on $X$; $\hat{X}$ represents the fitted matrix of $K$ exposures from (1) which equals $G\hat{C}$; $\beta$ expresses a $K$-dimensional vector that represents the causal effect of exposures on the outcome; $E$ is an $N$ by $M$ matrix of residual error; $\varepsilon$ is an $N$-vector of residual error.

Eq (1) regresses the exposure matrix with genotype matrix to predict the genetically fitted exposure matrix. In the above step, Partial Least Squares (PLS) regression is adopted to model this association, which accounts for the variation in $G$ and $X$ and the correlation between them [22]. Both $G$ and $X$ are assumed to be first scaled to have zero mean and unit variance for the respective column, and PLS regression is briefed as follows. First, the first component $\hat{u}_1 = G\rho^{(1)}$ and $\hat{v}_1 = X\gamma^{(1)}$ can be derived by solving the following conditional extremum problem, where $\rho^{(1)}$ denotes a $M$-vector loadings of SNPs and $\gamma^{(1)}$ is a $K$-vector loadings of exposures.

$$
\begin{aligned}
\max(\hat{u}_1^T \cdot \hat{v}_1) \quad &= \rho^{(1)T}G^T X\gamma^{(1)} \\
s.t. \ \rho^{(1)T}\rho^{(1)} \quad &= 1 \\
\gamma^{(1)T}\gamma^{(1)} \quad &= 1
\end{aligned}
\tag{3}
$$

The optimization problem can be solved using Lagrange multiplier or SVD decomposition. The solution equals to calculate the eigenvalue and eigenvectors of $G^T X X^T G$. Suppose $\theta = \rho^{(1)T}G^T X\gamma^{(1)}$, then the maximum eigenvalue of $G^T X X^T G$ is $\theta^2$. The corresponding unit eigenvector of $\theta^2$ is the solution $\rho^{(1)}$, and then $\gamma^{(1)}$ is calculated as $\frac{1}{\theta}X^T G\rho^{(1)}$. Second, the regression model of $X$ and $G$ on $\hat{u}_1$ is built, respectively.

$$
\begin{aligned}
G &= \hat{u}_1\sigma^{(1)T} + G_1^{res} \\
X &= \hat{u}_1\tau^{(1)T} + X_1^{res}
\end{aligned}
\tag{4}
$$

where $\sigma^{(1)} = G^T\hat{u}_1/||\hat{u}_1||^2$ and $\tau^{(1)} = X^T\hat{u}_1/||\hat{u}_1||^2$ denote the coefficient vector estimated using the ordinary least square method (OLS); $G_1^{res}$ and $X_1^{res}$ represent the residual matrix. Third, residual matrix $G_1^{res}$ and $X_1^{res}$ are substituted for $G$ and $X$, and the above steps (3)-(4) are repeated. Then we have the following parameter vectors: $\rho^{(2)}, \gamma^{(2)}, \hat{u}_2 = G\rho^{(2)}, \hat{v}_2 = X\gamma^{(2)}$, $\sigma^{(2)} = G^T\hat{u}_2/||\hat{u}_2||^2, \tau^{(2)} = X^T\hat{u}_2/||\hat{u}_2||^2$. The regression model after the second iteration is built as

$$
\begin{aligned}
G &= \hat{u}_1\sigma^{(1)T} + \hat{u}_2\sigma^{(2)T} + G_2^{res} \\
X &= \hat{u}_1\tau^{(1)T} + \hat{u}_2\tau^{(2)T} + X_2^{res}
\end{aligned}
\tag{5}
$$

$r$ repetitions are assumed to be conducted in total, and $r$ components $\hat{u}_1, \hat{u}_2, \ldots, \hat{u}_r$ are obtained. Subsequently, the following equations hold.

$$
\begin{aligned}
G &= \hat{u}_1\sigma^{(1)T} + \ldots + \hat{u}_r\sigma^{(r)T} + G_r^{res} \\
X &= \hat{u}_1\tau^{(1)T} + \ldots + \hat{u}_r\tau^{(r)T} + X_r^{res}
\end{aligned}
\tag{6}
$$

The regression coefficient $\hat{C}$ is calculated by taking each score $u_i = \rho_1^{(i)}g_1 + \ldots + \rho_M^{(i)}g_M$ ($1 \le i \le r$) back into $X = u_1\hat{\tau}^{(1)T} + \ldots + u_r\hat{\tau}^{(r)T}$ to fit the regression model $\hat{X} = G\hat{C}$. Suppose that after each iteration the vectors $\rho^{(i)}, \sigma^{(i)}, \tau^{(i)}$ are saved as columns in matrix $P, \Sigma, T$, then

the coefficient matrix $\hat{C}$ can be written in a less intuitive way as $\hat{C} = P(\Sigma^T P)^{-1} T^T$ [77]. To determine the optimal iteration times $r$ ($1 \leq r \leq \text{rank}(G)$), 10-fold CV was adopted to select the optimal tuning parameter $r$ that achieve the minimum cross-validation MSE. The PLS analysis was implemented using the R (version 4.2.1) package pls (version 2.8–1) [77]. The above coefficient matrix $\hat{C}$ and predicted exposure matrix $\hat{X}$ can be obtained through function *coef()* and *predict()*. There is no need to calculate all components to build the model. Accordingly, the largest iteration times was set as rank($X$) (usually equals $K$), which greatly reduces the computing time and make the linear combination of $G$ to explain the proportion of $X$ as much as possible.

Eq (2) regresses the outcome with the predicted exposure matrix derived from Eq (1) to estimate causal effect of the respective exposure on the outcome. The Lasso regression [23] is adopted to estimate the coefficient $\beta$ in this step by minimizing the following penalized loss function:

$$||Y - \hat{X}\beta||_2^2 + \lambda ||\beta||_1 \tag{7}$$

Grid Search and 10-fold CV were adopted to select the optimal tuning parameter $\lambda$ to achieve the minimum cross-validation MSE. The grid search range of $\lambda$ was set from $10^{-2}$ to $10^{10}$ (the exponent was uniformly taken 100 numbers from -2 to 10). The Lasso regression was implemented using the R package glmnet (version 4.1–4) [78]. The Lasso regression always applies to highly correlated variables, and enjoying favorable properties of subset selection and coefficient estimation. For better ranking exposures based on their strength of causality evidence, a regularized projection method known as de-sparsified Lasso was performed to estimate the *P*-values of the exposures [24]. The de-sparsified Lasso conducts a bias correction of the lasso estimator following an asymptotic normal distribution to derive *P*-values for the coefficient of each predictor (exposure). The de-sparsified Lasso method was employed using the R package hdi (version 0.1–9) [79]. The exposures with a non-zero coefficient and a significant *P*-value (e.g., lower than 0.05) were selected as causal risk factors.

The utilization of PLS regression in our method can be compared with previous MR approaches using techniques related to principal component analysis (PCA) [74,75]. In terms of algorithm, PLS regression is specifically designed for regression predictive tasks, such as the first stage of one-sample MR where we want to establish a relationship between instrumental and exposure variables. It integrates the dimensionality reduction process into its iterative regression steps between the input and response variables. In contrast, PCA is specifically designed for dimensionality reduction of input variables, while the response variables are not considered during the transformation process. In terms of interpretability, the use of PLS regression in our method does not affect the interpretability of MR, but the PCA-based approaches mentioned above does. The reason is that our method is designed within one-sample MR framework where the interpretability mainly lies in the second stage. Conversely, there is only one stage of linking SNP-outcome associations with SNP-exposure associations in two-sample MR where the PCA-based approaches are designed, thus grouping a set of original exposures into a new synthetic exposure through dimensionality reduction techniques will surely influence the interpretability.

## Winner's curse correction

In MR, a *P*-value threshold (usually $5 \times 10^{-8}$) is introduced as statistical significance to select SNPs serve as instruments from GWAS summary statistics of previous discovery data. The thresholding creates the phenomenon of Winner's Curse (WC), where the associations tend to be overestimated in the discovery population under the effect of chance noise [7]. WC is

identified when the identical sample for IV selection is employed for MR analysis. The assumption of MR suggests that the IVs should show a significant association with one of the exposures. On that basis, the selected candidate associations closed to the threshold are more likely to fail to reach the threshold, such that biased causal estimates in MR are generated. Using another independent sample to select IVs may avoid WC bias, whereas some limitations remain. For instance, with IDPs as exposures, due to the diversity of brain templates and imaging features and the population of Biobank data, few datasets are totally independent and have published GWAS with matched IDPs for MR analysis.

Thus, we seek to make correction for the SNP-exposure effect size. Previously, a computationally attractive WC-correction method termed Lasso-type WC-correction has been introduced and found to improve the predictive performance in PRS analysis [80]. We referred to the ideas from the method and applied it to MR setting. Each SNP and exposure are assumed to be first scaled to have a mean zero and a unit variance, and $\hat{\alpha}_{ik}$ is assumed as the marginal estimate of SNP $i$ with exposure $x_k$. The effect size of $M$ SNPs ($\alpha_{1k}, \alpha_{2k}, \ldots \alpha_{Mk}$) with exposure $x_k$ is estimated by minimizing the penalized loss function:

$$\sum_{i=1}^{N} \left( x_{ik} - \sum_{m=1}^{M} g_{im} \alpha_{mk} \right)^2 + \lambda \sum_{m=1}^{M} |\alpha_{mk}| \tag{8}$$

$\hat{\alpha}_{ik}^0$ is assumed as the marginal OLS estimate of SNP $i$ with exposure $k$. When the SNPs are independent, the solution to (8) is expressed as [22]:

$$\hat{\alpha}_{ik}^{corrected} = sign(\hat{\alpha}_{ik}^0)(|\hat{\alpha}_{ik}^0| - \gamma)^+, 1 \le i \le M, 1 \le k \le K \tag{9}$$

Eq (9) gives a shrinkage for the effect size of the SNP, which also equals

$$z_{ik}^{corrected} = sign(z_{ik}^0)(|z_{ik}^0| - \gamma)^+, 1 \le i \le M, 1 \le k \le K \tag{10}$$

where $z_{ik}$ denotes the $z$-score for the marginal estimator; $\gamma$ represents a constant in one-to-one correspondence to $\lambda$. A higher $\gamma$ gives stronger shrinkage of the SNP effect size. In MR, the selection of $\gamma$ conforms to two considerations. First, the selected instrumental SNP should show a significant association with the exposure. Accordingly, the shrinkage should be stronger when $P$-value threshold for IV selection is more relaxed. Second, a sufficiently large sample size is capable of reducing the uncertainty of the genetic associations while alleviating the bias arising from WC. Thus, the shrinkage should be more relaxed for larger sample sizes, which is expressed as:

$$\gamma = \frac{c}{-\log_{10}(threshold) + \log_{10}(N)}, \text{constant } c > 0 \tag{11}$$

In accordance with the description above, the WC-correction is conducted as follows:

Step 1. Given the initial set of candidate instrumental SNP-exposure pairs with $P$-values less than a predefined threshold, calculate the corresponding $z$-score through $|z| = \phi^{-1}(1 - p/2)$, where $\phi()$ is the density function of standard normal distribution;

Step 2. Perform shrinkage of the $z$-value for each SNP-exposure association:
$|z^{corrected}| = \left( |z| - \frac{c}{-\log_{10}(threshold)+\log_{10}(N)} \right)^+$;

Step 3. Calculate the corrected $P$-value corresponding to the shrank $z$-score: $p^{corrected} = 2 \times (1 - \phi(|z^{corrected}|))$. Thus, the SNP-exposure pairs with corrected $P$-values still less than the threshold will be retained for subsequent MR analyses. Notably, here we assume that the shrank $z$-score follows the standard normal distribution, thus enabling re-selecting instruments based on the corresponding pre-specified $P$-value threshold.

Note that the above steps can be also taken as a second selection of both instrumental SNPs and exposures (primarily SNPs, as there are usually a number of SNPs associated with the respective exposure). This approach only requires $P$-values from GWAS summary statistics, and is fast in computation for a number of SNP-exposure pairs and $P$-value thresholds. The choice of parameter $c$ is determined in simulation and discussed below.

## Detection of horizontal pleiotropy

MR assumes that all IVs affect the outcome only through the exposures (A3). If this assumption is violated, the MR estimate will be biased as there exists horizontal pleiotropy where genetic variants affect the outcome independently of exposures. To quantify the presence of pleiotropy, the Sargan test is employed, an established tool for evaluating pleiotropy in one-sample MR [81]. In brief, it tests whether IVs can explain the variation of the outcome which can't be explained by the exposures. Under the null hypothesis that no pleiotropy exists amongst the IVs and suppose there are more IVs than exposures ($M > K$), the Sargan test can be performed as follows:

Step 1. Calculate the residual term by $\boldsymbol{\varepsilon} = \boldsymbol{Y} - \hat{\boldsymbol{X}}\hat{\boldsymbol{\beta}}$, where $\hat{\beta}$ expresses the two-stage estimate of exposures on the outcome, $\hat{\boldsymbol{X}}$ denotes the predicted value of exposure matrix from the first stage;

Step 2. Regress the residual term on the full set of IVs: $\boldsymbol{\varepsilon} \sim \boldsymbol{G}$, and calculate $R^2$ of the model;

Step 3. Calculate the Sargan statistic by $S = N \cdot R^2$, and compare $S$ with the chi-square distribution $\chi^2(df = M - K)$. If the $P$-value of the Sargan statistic is less than 0.05, the null hypothesis will be rejected. Subsequently, the result of this MR analysis turns out to be unreliable and should be discarded.

## Simulation analyses

Simulations were performed to evaluate the performance of MR-PL and compare it with other approaches. For baseline simulation, $m = 5,000$ total number of SNPs as potential instruments, $K = 20$ total number of exposures, and one outcome for $N = 10,000$ independent samples were generated using the following procedure. First, each SNP was independently drawn from binomial distribution B(2, 0.3), where the minor allele frequency of a SNP was set to 0.3. The genotype for each SNP was scaled to have a zero mean and a unit variance. For the respective exposure, causal SNPs were randomly selected from the whole SNP set with the probability $\pi$. $\boldsymbol{\gamma}_{x_k}$ is denoted as the $m$-vector of SNP effect size on the exposure $x_k$, which was generated from normal distribution $N(0, \frac{hg^2}{\pi \cdot m})$ for those causal SNPs, and was set to zero for those non-causal SNPs, where $hg^2$ was proportion of variance in the respective exposure explained by the SNPs. Next, exposures and the outcome were simulated based on the standardized genotype matrix $\boldsymbol{G}$ and the vector $\gamma_{x_k}$ as follows:

$$\boldsymbol{X}_k = \boldsymbol{G}\gamma_{x_k} + \sum\nolimits_{i=1, i \neq k}^{K} \alpha_{u_i x} \boldsymbol{U}_i + \boldsymbol{\varepsilon}_{x_k} \tag{12}$$

$$\boldsymbol{Y} = \sum\nolimits_{i=1}^{K} \beta_{x_i y} \boldsymbol{X}_i + \sum\nolimits_{i=1}^{K} \delta_{u_i y} \boldsymbol{U}_i + \boldsymbol{\varepsilon}_y \tag{13}$$

where the $N$-vector residual errors $\boldsymbol{\varepsilon}_{x_k}$ ($1 \leq k \leq K$) and $\boldsymbol{\varepsilon}_y$ were independently drawn from $N(0, \sigma^2)$, $\sigma = 0.1$; $\alpha_{u_k x}$ and $\delta_{u_k y}$ denote the effect size of confounder $u_k$ on exposure $x_k$ and the outcome $y$, respectively; $\boldsymbol{U}_k$ denotes the $N$-vector observations of confounder $u_k$ ($1 \leq k \leq K$) and

was independently drawn from $N(0, (1 - hg^2 - \sigma^2)/\sum_{i=1,i\neq k}^{K} \alpha_{u_i x}^2)$, leading to correlations among exposures and the outcome; $\beta_{x_k y}$ expresses the causal effect size of exposure $x_k$ and the outcome $y$. $\boldsymbol{Y}$ was scaled to have a zero mean and unit variance. The above simulation ensures that standardized genotypes, exposures, and the outcome served as the input for MR analysis.

For the parameter setting, $hg^2$ was set as 0.2, 0.3, 0.4 or 0.5 as the heritability of most IDPs were reported to fall around this range [30]. For $\pi$, it was set as 0.05 with the respective exposure having large polygenicity. For the effect of confounding factors on exposures $\alpha_{u_k x}$ ($1 \leq k \leq K$), it was randomly drawn from uniform distribution of either U(0,5) or an increased effect size U(5,10). For the effect of confounding factors on the outcome $\delta_{u_k y}$, it was randomly drawn from uniform distribution U(0,1). For the true causal effect of exposures $\beta_{x_k y}$ ($1 \leq k \leq K$), it was randomly drawn from discrete sets of {-0.1,0,0.1}, {-0.2,0,0.2} or {-0.3,0,0.3}, or continuous set U(-0.1,0.1), U(-0.2,0.2) or U(-0.3,0.3). A total of 48 parameter settings were designed, which is summarized in S1 Table. For each parameter setting, we generated 100 replications of simulation datasets. To better mimic real MR application, a GWAS was performed for the respective exposure in the respective replication of simulated dataset. For MR-PL and all competing approaches described below, the SNPs associated with at least one exposure at a marginal $P$-value below $5\times10^{-8}$ were selected as the initial instruments. The above parameter settings presented the desired numbers of instrumental SNPs ranging from 256 to 1,092, moderate to high correlation between any two exposures ranging from 0.31 to 0.72, and overlapping percentage of causal or instrumental SNPs between any two exposures ranging from 0% to 18.18% (S23 Table). To access performance, the MSE and type I error (at 5% nominal level) were calculated for each parameter setting (averaged across 100 replications).

For the simulation of WC-correction, the above simulation was expanded to cover different strength of $P$-value thresholds for instrument selection. $P$-value thresholds were set to be seven values of $5\times10^{-4}$, $5\times10^{-5}$, $5\times10^{-6}$, $5\times10^{-7}$, $5\times10^{-8}$, $5\times10^{-9}$ or $5\times10^{-10}$. For the tuning parameter $c$ in Eq (11), it varied from 5 to 40 (step by 5). A higher $c$ gives stronger WC-correction and less SNPs are retained as instruments. For the choice of $c$, the index $\Delta$MSE was defined, where $\Delta$MSE denotes the percentage decrease in MSE from MR analysis without WC-correction to MR analysis with WC-correction. The index $\Delta$MSE was then averaged across all 48 parameter settings, and the optimal $c$ was selected with the highest $\Delta$MSE.

Besides, the simulation study was also expanded to different correlation structure between instrumental SNPs, and scenarios with the presence of pleiotropy. More details are presented in the S1 Text.

### Comparison with other methods

We compared the performance of MR-PL in the simulated dataset with seven other approaches that include the following:

(1) Two-step least-squared (2SLS) based methods that include: multivariable 2SLS (Multi-2SLS), univariable 2SLS (Uni-2SLS), polygenic risk score-based 2SLS (PRS-2SLS), and MV-IWAS. In Multi-2SLS, the respective exposure is first regressed on the instrumental matrix through multivariable linear regression, and the outcome is then regressed on the fitted values of all exposures from the first stage through multivariable linear regression [19]. For the instrumental matrix $\boldsymbol{G}$, exposure matrix $\boldsymbol{X}$ and the outcome $\boldsymbol{Y}$, the causal effect estimate of Multi-2SLS can be derived by:

$$\boldsymbol{\beta} = (\beta_1, \beta_2, \ldots, \beta_K) = [\boldsymbol{X}^T \boldsymbol{G} (\boldsymbol{G}^T \boldsymbol{G})^{-1} \boldsymbol{G}^T \boldsymbol{X}]^{-1} \boldsymbol{X}^T \boldsymbol{G} (\boldsymbol{G}^T \boldsymbol{G})^{-1} \boldsymbol{G}^T \boldsymbol{Y} \tag{14}$$

Similar to Multi-2SLS, the causal estimate of Uni-2SLS can be derived by first regressing the exposure on the instrumental matrix through multivariable linear regression, and then regress the outcome on the fitted value of the exposure through univariable linear regression [82]. $\boldsymbol{G}_{x_k}$ is assumed as the instrumental matrix corresponding to variants that show a significant association with exposure $x_k$, and $\boldsymbol{X}_k$ is the $N$-vector observations corresponding to $x_k$, the causal effect estimate of Uni-2SLS can be derived by:

$$\beta_k = [\boldsymbol{X}_k{}^T\boldsymbol{G}_{x_k}(\boldsymbol{G}_{x_k}{}^T\boldsymbol{G}_{x_k})^{-1}\boldsymbol{G}_{x_k}{}^T\boldsymbol{X}_k]^{-1}\boldsymbol{X}_k{}^T\boldsymbol{G}_{x_k}(\boldsymbol{G}_{x_k}{}^T\boldsymbol{G}_{x_k})^{-1}\boldsymbol{G}_{x_k}{}^T\boldsymbol{Y} \qquad (15)$$

In PRS-2SLS, a polygenic risk score is served as the instrument, which is a weighted or unweighted mean over the genotypes of multiple SNPs [82,83]. In the real application, it's usually hard to obtain an independent and phenotype-matched dataset to determine the weight assigned to each SNP. Thus, we calculated the unweighted mean over the genotypes of instrumental SNPs and use it as a single instrument in 2SLS procedure. The causal effect estimate of PRS-2SLS can be derived by replacing $\boldsymbol{G}_{x_k}$ in Eq (15) with its row mean. For the above three 2SLS-based methods, we used function *tsls()* in R package *sem* (version 3.1–15) (http://cran.r-project.org/web/packages/sem/index.html) to implement the 2SLS procedure. MV-IWAS is a comprehensive multivariable 2SLS-based MR method which only used SNPs specific to each exposure as IVs [25]. Therefore, we tried this implementation in our one-sample simulation, only using the exposure-specific IVs for the prediction of each exposure in the first stage of 2SLS.

(2) ImagingMR, a MR framework used to jointly select IVs and exposures in a one-sample MR setting [20]. Given a candidate set of instrumental SNPs and exposures, ImagingMR first jointly selects subset of IVs and exposures through bi-clustering procedure, and then transforms the selected exposures into a synthetic exposure and estimates the causal effect of the new synthetic exposure on the outcome. Note that ImagingMR cannot directly estimate the causal effect of individual exposure, but it can be derived by reducing the synthetic exposure to original exposures. Suppose $X^* = (X_{M_1}, \ldots, X_{M_p})$ is the exposure matrix of $p$ selected exposures after jointly selection of instruments and exposures, and $\boldsymbol{M}^* = X^* \cdot \boldsymbol{\varphi} = (\boldsymbol{M}_1, \ldots, \boldsymbol{M}_V)$ is the orthogonally transformed matrix of $X^*$, where $\boldsymbol{\varphi} = (\boldsymbol{\varphi}_1, \ldots, \boldsymbol{\varphi}_V)$ is the $p$ by $V$ transforming matrix. ImagingMR estimates the causal effect $\theta_v$ of synthetic exposure $\boldsymbol{M}_v$ ($1 \leq v \leq V$) through IVW method [82]. Then the causal estimate of original exposures can be extracted from $\hat{Y} = \sum_{v=1}^V \theta_v \boldsymbol{M}_v = \sum_{v=1}^V \theta_v X^* \varphi_v$. We implemented ImagingMR using codes on https://github.com/kehongjie/ImagingMR.

(3) We also replaced the second stage of MR-PL (Lasso regression) with other frequently used regularized methods including Ridge regression [84], Elastic-net regression [85], and Least-angle regression (Lars) [86], denoted as PLS-Ridge, PLS-Elasticnet and PLS-Lars, respectively. For the above-described regularized methods, a 10-fold CV was used to select the optimal tuning parameter(s) to achieve the minimum MSE. We implemented Ridge and Elasticnet regression using R package *glmnet* (version 4.1–4) [78], and Lars regression using R package *lars* (version 1.3) [87].

## Application on real datasets

We applied MR-PL to test the putative causal effect of white matter microstructure on 180 biological and lifestyle traits in the UKBB data. We included 36 diffusion tensor IDPs in the analysis. The white matter tracts are segmented by JHU-label atlas [88,89], and the microstructure

of the above-mentioned tracts are measured by the mean fractional anisotropy (FA) (S7 Table). As shown in S27 Fig, The IDPs demonstrated moderate to large correlation with each other. For IV selection, the GWAS summarized data with 33,292 British samples from the UKBB provided by Zhao et al. were used [28]. Since Zhao et al. took the mean of FA measures for any left/right tract pairs rather than making a distinction between tracts in the left and right hemisphere, there were 20 individual white matter tracts in Zhao et al. (S9 Table) corresponding to 36 ones in the UKBB (S7 Table). The global DTI measure of the mean FA value averaged across all tracts was also included for IV selection. Finally, 368 independent genetic variants ($r^2 < 0.1$, S9 Table) served as the initial IVs, which were reported to be associated with the mean FA value of at least one white matter tract at genome-wide significance level ($5 \times 10^{-8}$) from Zhao et al. The sample quality control was performed with the following steps: 1) removal of non-British samples (Field ID 21000); 2) removal of samples with poor heterozygosity or missingness (Field ID 22010); 3) removal of genetic relatedness by excluding individuals randomly in a pair of samples estimated to be genetically related (Field ID 22011); 4) removal of samples whose genetic sex is inconsistent with report sex (Field ID 31/22001); 5) removal of samples without imaging data. After that, 32,667 European samples were left with matched genotype and imaging data. The 180 complex traits included as outcomes were all continuous and manually divided into ten categories following the previous literature [90], which comprised 65 biochemical measures, 24 physical measures, 19 cognitive traits, 14 health related traits, 17 eating behaviors, 10 lifestyle traits, 13 alcohol intake behaviors, eight smoking behaviors, seven mental health related traits, as well as three socioeconomic measures. For the respective trait, individuals with missing values were directly removed from the analysis. After aligning each of the complex traits with the genotype and imaging data in turn, the MR analysis was limited to those with at least 2000 individuals to ensure sufficient statistical power. Finally, the sample size retained for MR analysis ranged from 2087 to 32,666. More detailed information is presented in S8 Table and S25 Fig.

Before MR analysis, each IDP was regressed upon various covariates provided by the UKBB, comprising age, $age^2$, sex, age-sex interaction, $age^2$-sex interaction, genotype measurement batch, assessment site, BMI, as well as the top 40 genetic principle components. The obtained IDP residuals, as well as the instrumental SNPs and outcome traits were standardized to have a mean zero and unit variance, and were finally employed for MR analysis. Since the same sample for IV selection was used for MR analysis, WC-correction was also performed. Furthermore, the F-statistics was calculated to test whether the instrumental SNPs jointly strongly predict each of the exposures. Besides the main analysis with independent variants as instruments, parallel analysis was conducted with 758 variants in linkage disequilibrium (LD, $r^2 < 0.6$, S15 Table), derived from the same GWAS on brain white matter [28]. For traits categorized in biochemical and physical measures, sensitivity analysis was conducted by adding additional potential confounding factors as covariates, including LDL, HDL cholesterol, triglycerides, alcohol intake frequency and pack years of smoking, which have been reported to be associated with blood cell composition and physical measurements [91–95].

## Prediction of causal SNP-IDP-trait pathways

To gain more insights into the genetic basis underlying the causal relationship between white matter microstructure and complex traits, a leave-one-out CV was performed to estimate the significance of SNPs with IDPs at the first stage of MR-PL (PLS regression). Specifically, the leave-one-out CV is a special form of *N*-fold CV, where *N* equals to the number of samples in the dataset. Following the implementation of leave-one-out CV in PLS regression, the coefficient variance and significance of SNPs with IDPs were derived through the jackknife

approximate *t* test [96], which was implemented through *jack.test()* function within the R package pls (version 2.8–1). The SNP-white matter tract-trait pathway was constructed if it satisfied the following requirements: (1) the SNP showed a significant association with the white matter tract at a Bonferroni-corrected *P*-value threshold of $2.13 \times 10^{-5}$ (0.05/69/34); (2) the white matter tract was associated with the trait in MR-PL ($\beta \neq 0$, $P < 0.05$). Furthermore, thorough searches were conducted in the NHGRI-EBI GWAS Catalog (version 2022-09-30, www.ebi.ac.uk/gwas/) [64] to find previously reported associations of all the genes within 10 kb of the instrumental SNPs.

## Supporting information

**S1 Fig. The data generation framework in baseline simulation.**
(PDF)

**S2 Fig. Bias and variance results of MR-PL and other MR approaches in baseline simulation.** Bias was calculated as the mean of the absolute difference between causal estimate and true effect, and variance was calculated as the variance of causal estimate minus true effect (averaged across 100 replications for each setting). The results from the methods of ImagingMR and PRS-2SLS were not included here. The full results can be found in S3 Table.
(PDF)

**S3 Fig. Type I error rate of MR-PL and other MR approaches in supplementary simulation with the presence of linkage disequilibrium.** Simulation settings were included if the causal effect was drawn from the discrete set. MR approaches were included if individual *P*-value of causal effect estimate of each exposure can be calculated. A method fails to control type I error if its type I error rate exceeds the nominal significance level of 0.05 (red dotted line). The error bar represents the variance of type I error rate over 100 replications. For settings with too small variance, the error bar tends to degenerate to a point.
(PDF)

**S4 Fig. Type I error rate of MR-PL and other MR approaches in supplementary simulation with the presence of pleiotropy.** Simulation settings were included if the causal effect was drawn from the discrete set. A method fails to control type I error if its type I error rate exceeds the nominal significance level of 0.05 (red dotted line). The error bar represents the variance of type I error rate over 100 replications.
(PDF)

**S5 Fig. Sargan test *P*-value distribution of MR-PL in baseline simulation where no pleiotropy exists.** A histogram illustrating the distribution of *P*-values across 100 replications is presented for each parameter setting. A *P*-value exceeding 0.05 indicates the absence of pleiotropy.
(PDF)

**S6 Fig. Sargan test *P*-value distribution of MR-PL in supplementary simulation with the presence of pleiotropy.** A histogram illustrating the distribution of *P*-values across 100 replications is presented for each parameter setting. A *P*-value below 0.05 indicates the presence of pleiotropy.
(PDF)

**S7 Fig. Sargan test *P*-value distribution of MR-PL with WC-correction in baseline simulation where no pleiotropy exists.** A histogram illustrating the distribution of *P*-values across 100 replications is presented for each parameter setting. A *P*-value exceeding 0.05 indicates the

absence of pleiotropy.
(PDF)

**S8 Fig. Sargan test *P*-value distribution of MR-PL with WC-correction in supplementary simulation with the presence of pleiotropy.** A histogram illustrating the distribution of *P*-values across 100 replications is presented for each parameter setting. A *P*-value below 0.05 indicates the presence of pleiotropy.
(PDF)

**S9 Fig. Power results of MR-PL and other MR approaches in baseline simulation.** Simulation settings were included if the causal effect was drawn from the discrete sets. The error bar represents the variance of power over 100 replications in each parameter setting. For settings with too small variance, the error bar tends to degenerate to a point.
(PDF)

**S10 Fig. A comparison of mean squared error (MSE) between MR-PL with and without winner's curse correction (WC-correction) at *c* = 15 in baseline simulation.** The error bar represents ten-fold variance in MSE over 100 replications in each parameter setting. For settings with too small variance, the error bar tends to degenerate to a point.
(PDF)

**S11 Fig. A comparison of mean squared error (MSE) between MR-PL with and without winner's curse correction (WC-correction) at *c* = 25 in baseline simulation.** The error bar represents ten-fold variance in MSE over 100 replications in each parameter setting. For settings with too small variance, the error bar tends to degenerate to a point.
(PDF)

**S12 Fig. A comparison of type I error rate between MR-PL with and without winner's curse correction (WC-correction) at *c* = 15 in baseline simulation.** Simulation settings were included if the causal effect was drawn from the discrete set. The error bar represents the variance of type I error rate across 100 replications for each parameter setting. For settings with too small variance, the error bar tends to degenerate to a point.
(PDF)

**S13 Fig. A comparison of type I error rate between MR-PL with and without winner's curse correction (WC-correction) at *c* = 25 in baseline simulation.** Simulation settings were included if the causal effect was drawn from the discrete set. The error bar represents the variance of type I error rate across 100 replications for each parameter setting. For settings with too small variance, the error bar tends to degenerate to a point.
(PDF)

**S14 Fig. A comparison of power results between MR-PL with and without winner's curse correction (WC-correction) at *c* = 15 (and other approaches) in baseline simulation.** Simulation settings were included if the causal effect was drawn from the discrete set. The error bar represents the variance of power across 100 replications for each parameter setting. For settings with too small variance, the error bar tends to degenerate to a point.
(PDF)

**S15 Fig. A comparison of power results between MR-PL with and without winner's curse correction (WC-correction) at *c* = 25 (and other approaches) in baseline simulation.** Simulation settings were included if the causal effect was drawn from the discrete set. The error bar represents the variance of power across 100 replications for each parameter setting. For

settings with too small variance, the error bar tends to degenerate to a point.
(PDF)

**S16 Fig. A comparison of mean squared error (MSE) between MR-PL with and without winner's curse correction (WC-correction) across different $P$-value thresholds to select instrumental variants in baseline simulation.** The error bar represents ten-fold variance in MSE over 100 replications in each parameter setting. For settings with too small variance, the error bar tends to degenerate to a point.
(PDF)

**S17 Fig. A comparison of type I error rate between MR-PL with and without winner's curse correction (WC-correction) across different $P$-value thresholds to select instrumental variants in baseline simulation.** Simulation settings are included if the causal effect was drawn from the discrete set. The error bar represents the variance of type I error rate across 100 replications for each parameter setting. For settings with too small variance, the error bar tends to degenerate to a point.
(PDF)

**S18 Fig. A comparison of mean squared error (MSE) between MR-PL with and without winner's curse correction (WC-correction) at $c = 15$ across different $P$-value thresholds to select instrumental variants in baseline simulation.** The error bar represents ten-fold variance in MSE over 100 replications in each parameter setting. For settings with too small variance, the error bar tends to degenerate to a point.
(PDF)

**S19 Fig. A comparison of mean squared error (MSE) between MR-PL with and without winner's curse correction (WC-correction) at $c = 25$ across different $P$-value thresholds to select instrumental variants in baseline simulation.** The error bar represents ten-fold variance in MSE over 100 replications in each parameter setting. For settings with too small variance, the error bar tends to degenerate to a point.
(PDF)

**S20 Fig. A comparison of type I error rate between MR-PL with and without winner's curse correction (WC-correction) at $c = 15$ across different $P$-value thresholds to select instrumental variants in baseline simulation.** Simulation settings were included if the causal effect was drawn from the discrete set. The error bar represents the variance of type I error rate across 100 replications for each parameter setting. For settings with too small variance, the error bar tends to degenerate to a point.
(PDF)

**S21 Fig. A comparison of type I error rate between MR-PL with and without winner's curse correction (WC-correction) at $c = 25$ across different $P$-value thresholds to select instrumental variants in baseline simulation.** Simulation settings were included if the causal effect was drawn from the discrete set. The error bar represents the variance of type I error rate across 100 replications for each parameter setting. For settings with too small variance, the error bar tends to degenerate to a point.
(PDF)

**S22 Fig. A comparison of mean squared error (MSE) between MR-PL with and without winner's curse correction (WC-correction) across different $P$-value thresholds to select instrumental variants in supplementary simulation with the presence of linkage disequilibrium.** The error bar represents ten-fold variance in the MSE over 100 replications in each

parameter setting. For settings with too small variance, the error bar tends to degenerate to a point.
(PDF)

**S23 Fig. A comparison of type I error rate between MR-PL with and without winner's curse correction (WC-correction) across different *P*-value thresholds to select instrumental variants in supplementary simulation with the presence of linkage disequilibrium.** Simulation settings were included if the causal effect was drawn from the discrete set. The error bar represents the variance of type I error across 100 replications for each parameter setting. For settings with too small variance, the error bar tends to degenerate to a point.
(PDF)

**S24 Fig. Results of conditional F-statistic values ($F_{sw}$) in baseline simulation. (A)** Boxplot of the conditional F-statistics before and after WC-correction, with a mean of 4.37 before WC-correction, and 5.63 after WC-correction. **(B, D)** Scatter plots colored by density of the conditional F-statistics and exposure numbers across 43,200 simulations (100 replications × 48 settings × 9 *c* values), where **(D)** displays separate plots for different parameter settings of heritability. **(C, E)** Scatter plots colored by density of conditional F-statistics and instrumental SNP numbers across 43,200 simulations (100 replications × 48 settings × 9 *c* values), where **(E)** displays separate plots for different parameter settings of heritability. **(F)** A comparison of Mean squared error (MSE) for MR-PL across different bins of conditional F-statistic values. The error bar represents the variance of MSE. **(E)** A comparison of type I error rate for MR-PL across different bins of conditional F-statistic values. The error bar represents the variance of type I error rate. The red dotted line denotes the rule-of-thumb value of 10. *r*: the estimate of Spearman correlation; *P*: the *P*-value of Spearman correlation test; n_neighbors: the number of dots around each dot.
(PDF)

**S25 Fig. Total number of traits in each trait categories from the UK Biobank.**
(PDF)

**S26 Fig. Percentage of non-causal SNPs selected as IVs before and after WC-correction.** Each dot represents the percentage of non-causal SNPs selected as IVs under a specific parameter setting (number of non-causal SNPs selected as instruments / total number of non-causal SNPs), averaged across 100 replications. The Wilcoxon signed rank test was used for testing the difference of the percentage before and after WC-correction across all 48 parameter settings. It showed that the WC-correction procedure could significantly reduce the chances for non-causal SNPs chosen as IVs which would otherwise bias MR results greatly.
(PDF)

**S27 Fig. Correlation between different white matter tracts. (A)** Correlation between the mean fractional anisotropy (FA) of white matter tracts before regressing out the covariates. **(B)** Correlation between the mean FA value of white matter tracts after regressing out the covariates. The complete names of these white matter tracts that correspond to the abbreviation used can be found in S7 Table.
(PDF)

**S1 Table. Different parameter settings under different scenarios in simulation.**
(XLSX)

**S2 Table. Mean squared error (MSE) of MR-PL and other MR approaches in baseline simulation. For each parameter setting, MSE was averaged over 100 replications**. The lowest

MSE is highlighted in bold in each scenario, and the second lowest MSE is highlighted in underline.
(XLSX)

**S3 Table. Bias and variance results (values within parentheses) of MR-PL and other MR approaches in baseline simulation.** Bias was calculated as the mean of the absolute difference between causal estimate and true effect, and variance was calculated as the variance of causal estimate minus true effect (averaged across 100 replications for each setting). The lowest value is highlighted in bold in each scenario, and the second lowest value is highlighted in underline.
(XLSX)

**S4 Table. Mean squared error (MSE) of MR-PL and other MR approaches in supplementary simulation with the presence of linkage disequilibrium.** For each parameter setting, MSE was averaged over 100 replications. The lowest MSE is highlighted in bold in each scenario, and the second lowest MSE is highlighted in underline.
(XLSX)

**S5 Table. Mean squared error (MSE) of MR-PL and other MR approaches in supplementary simulation with the presence of pleiotropy.** For each parameter setting, MSE was averaged over 100 replications. The lowest MSE is highlighted in bold in each scenario, and the second lowest MSE is highlighted in underline.
(XLSX)

**S6 Table. Conditional F-statistics of exposures under different parameter settings in baseline simulation.** Details for the calculation of the conditional F-statistic can be found in S1 Text.
(XLSX)

**S7 Table. The imaging-derived phenotypes (IDPs) used in this research from the UK Biobank.** The white matter tracts are measured by the mean fractional anisotropy (FA), and are segmented by JHU-label atlas. FieldID: the corresponding field ID information for these IDPs in the UK Biobank.
(XLSX)

**S8 Table. Detailed information of 180 complex traits used in this research from the UK Biobank.** N_full: the total sample size of this trait in the UK Biobank. N_matched: the sample size of this trait after the alignment with the genotype and brain imaging data. Coding: the UK Biobank coding identifier. Missing: the values encoded as missing for the trait. Mapping: the json-object describing the mapping used.
(XLSX)

**S9 Table. Independent variants used as initial instrumental variables for MR analysis.** The variants ($r^2 < 0.1$) are selected from Zhao, et al (PMID: 34140357), which have been reported to be associated with at least one white matter tract (measured by the mean fractional anisotropy, FA) at 5E-8 GWAS significance level. The term 'Average' refers to the mean FA value averaged across all tracts. It is a global dMRI measure of the whole brain. rsID: rsID of the SNP. chr: chromosome. pos: position on hg19. WM: the white matter tract reported to be associated with the corresponding SNP in Zhao, et al. P: the GWAS P-value reported in Zhao, et al.
(XLSX)

**S10 Table. F-statistics for imaging-derived phenotypes (IDPs).** Briefly, the F-statistic is derived as $R^2 \times (N\text{-}M\text{-}1) / (1\text{-}R^2) \times M$, where $R^2$ denotes the explained variance of instrumental SNPs, $N$ denotes the sample size, and $M$ denotes the total number of instrumental SNPs. The

instrumental SNPs for F-statistic computation are the ones kept after the winner's curse correction in MR-PL. Besides, the two IDPs (Field ID 25062 and 25063) that were discarded after the winner's curse correction due to a lack of associated instrumental SNPs are labeled as red, which happened to have F-statistics less than 10. FieldID: the corresponding field ID information for these IDPs in the UK Biobank.
(XLSX)

**S11 Table. Conditional F-statistics for imaging-derived phenotypes (IDPs).** Details for the calculation of the conditional F-statistic can be found in S1 Text. FieldID: the corresponding field ID information for these IDPs in the UK Biobank.
(XLSX)

**S12 Table. Summary of MR results for pairs of imaging-derived phenotypes (IDPs) and complex traits having a non-zero causal effect estimate.**
(XLSX)

**S13 Table. Previous evidence supporting a relationship between pairs of imaging-derived phenotypes (IDPs) and complex traits shown in Table 1, which are indicated by a non-zero causal effect estimate at *P*-value < 0.05.**
(XLSX)

**S14 Table. Supplementary MR results for the 9 IDP-trait pairs presented in Table 1, wherein the traits were categorized in biochemical and physical measures, after adding additional potential confounding factors as covariates.** The additionally added covariates includes LDL (Field ID 30780), HDL cholesterol (Field ID 30760), triglycerides (Field ID 30870), alcohol intake frequency (Field ID 1558), and pack years of smoking (Field ID 20161).
(XLSX)

**S15 Table. Correlated variants used as initial instrumental variables for supplementary MR analysis.** The variants ($r^2 < 0.6$) are selected from Zhao, et al (PMID: 34140357), which have been reported to be associated with at least one white matter tract (measured by measured by the mean fractional anisotropy, FA) at 5E-8 GWAS significance level. rsID: rsID of the SNP. chr: chromosome. pos: position on hg19. WM: the white matter tract reported to be associated with the SNP in Zhao, et al. *P*: the GWAS *P*-value reported in Zhao, et al.
(XLSX)

**S16 Table. Summary of supplementary MR results with correlated SNPs ($r^2 < 0.6$) as instruments for pairs of imaging-derived phenotypes (IDPs) and complex traits having a non-zero causal effect estimate.**
(XLSX)

**S17 Table. Summary of supplementary MR-PL results with winner's curse correction (WC-correction) at *c* = 15 for pairs of imaging-derived phenotypes (IDPs) and complex traits having a non-zero causal effect estimate.**
(XLSX)

**S18 Table. Summary of supplementary MR-PL results with winner's curse correction (WC-correction) at *c* = 25 for pairs of imaging-derived phenotypes (IDPs) and complex traits having a non-zero causal effect estimate.**
(XLSX)

**S19 Table. Summary of supplementary MR-PL results with no winner's curse correction (WC-correction) applied for pairs of imaging-derived phenotypes (IDPs) and complex**

**traits having a non-zero causal effect estimate.**
(XLSX)

**S20 Table. Summary of MR results (instrumental SNPs with $r^2 < 0.1$) for 2SLS-based methods of Uni-2SLS and Multi-2SLS.**
(XLSX)

**S21 Table. Summary of the 224 putative causal SNP-IDP-trait pathways.** The identified pathways are constructed by integrating 435 significant SNP-IDP pairs ($P$-value < 2.13E-05) and 26 IDP-trait pairs (a non-zero causal effect estimate at $P$-value < 0.05).
(XLSX)

**S22 Table. The mapped genes and their identified GWAS catalog traits (version 2022-09-30, www.ebi.ac.uk/gwas/) for instrumental SNPs in the 224 putative causal SNP-IDP-trait pathways.**
(XLSX)

**S23 Table. The number of instrumental variables (IVs), correlation between exposures, and overlapping percentage of causal SNPs and IVs between exposures under different scenarios in simulation.**
(XLSX)

**S1 Text. Supplementary methods.**
(PDF)

## Author Contributions

**Conceptualization:** Yucheng T. Yang, Xing-Ming Zhao.

**Formal analysis:** Anyi Yang.

**Funding acquisition:** Yucheng T. Yang, Xing-Ming Zhao.

**Investigation:** Anyi Yang.

**Methodology:** Anyi Yang.

**Project administration:** Yucheng T. Yang.

**Resources:** Xing-Ming Zhao.

**Software:** Anyi Yang.

**Supervision:** Yucheng T. Yang, Xing-Ming Zhao.

**Validation:** Anyi Yang, Yucheng T. Yang, Xing-Ming Zhao.

**Visualization:** Anyi Yang.

**Writing – original draft:** Anyi Yang.

**Writing – review & editing:** Yucheng T. Yang, Xing-Ming Zhao.

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
