## [Decision Letter · Decision Letter 0]

15 Jul 2023

Dear Dr Zhao,

Thank you very much for submitting your Research Article entitled 'An augmented Mendelian randomization approach provides causality of brain imaging features on complex traits in a single biobank-scale dataset' to PLOS Genetics.

The manuscript was fully evaluated at the editorial level and by independent peer reviewers. The reviewers appreciated the attention to an important problem, but raised some substantial concerns about the current manuscript. Based on the reviews, we will not be able to accept this version of the manuscript, but we would be willing to review a much-revised version. We cannot, of course, promise publication at that time.

If you decide to revise the manuscript for further consideration at PLOS Genetics, please aim to resubmit within the next 60 days, unless it will take extra time to address the concerns of the reviewers, in which case we would appreciate an expected resubmission date by email to plosgenetics@plos.org.

We are sorry that we cannot be more positive about your manuscript at this stage. Please do not hesitate to contact us if you have any concerns or questions.

Yours sincerely,

Zoltán Kutalik, PhD

Academic Editor

PLOS Genetics

Hua Tang

Section Editor

PLOS Genetics

As the authors can see from the reviewers comments, while the topic, the method and the application are of general interest, there are substantial improvements and clarifications necessary to be considered for publication in PLoS Genetics.

Reviewer's Responses to Questions

**Comments to the Authors:**

Reviewer #1: This is a useful look into partial least squares as a method in the MVMR framework, in cases of extreme upscaling

of the number of exposures. Overall, I appreciate the motivation and the applied results and I would recommend

that the methods and the simulation study be enhanced and contextualised in related MR literatured.

Introduction

-Thinking of the MR assumptions, the central issue at performing a multivariable

MR without any further methodological refining is the issue of conditionally weak

instruments (10.1002/sim.9133). As you correctly point out, a genetic variant is theoretically

affecting many brain regions/densities rather than only a single one, and the

result of this in the MVMR framework would be most pronounced by low conditional instrument

strength. Overall, I think the simulations studies could track this explicitly as there is

now

-While the application to imaging data is interesting and imaging data is a

good contender for highly correlated data, there is discussion on MR literature

on whether the target of the estimations is a modifiable exposure. Brain volumes

are not readily modifiable in the adult life,

-There is previous work tackling a similar issue with PCA and sparse PCA (10.7554/eLife.80063), as well

as applied work on components of body composition (DOI: 10.1038/s42003-021-02550-y).

The common thread in these works and your work is an improvement in precision with

a less clear result on bias. I would encourage you to present separate simulation

results for bias and

Results

-How did you choose the performance measure of MSE? Granted it is a combination

of bias and variance, but I've found it difficult to tell exactly where the benefit

of MR-PL lies, in bias or in variance? I imagine it is predominantly due to a reduction

in variance of the obtained estimates, similar to two other approaches mentioned above

-It would be interesting to see the results of the method but the results of

a simple two-stage least squares approach, univariable and multivariable, and

whether the observed differences in the real data examples line up with the

simulation results. I would expect that there will be less strong effects

in MVMR given the conditionally weak instrument problems.

-The WC-corrected version of the method appears to provide lower MSE. How do

you interpret this? In my understanding, a WCC would be targeting erroneously precise

estimates that have just crossed the limit of selection. I would think that these

SNPs should be still included, despite their increased variance when corrected.

Methods

-If I understand correctly, PLS is used and the X-Y covariance-explaining

components that are generated are fixed to be equal to the number of exposures.

Therefore, it is not primarily used as a dimensionality reduction approach.

In PCA approaches, including those described in the above two papers for MR,

the target is to generate components that explain the variance of X , and then

use those in second-stage causal analyses. Could you discuss this?

-In the data generating mechanism for the exposure, it is not clear how exactly the MVMR

assumptions would fail. Shouldn't there be a lack of exposure-specific instruments? It's

not clear to me how, out of a pool of 5,000 SNPs, of whom a random selection is affecting

each exposure, would violate the assumption that there is sufficient conditional instrument

strength. That is, I imagine MVMR might be feasible if, of those 5,000 SNPs, some are affecting

X_1, some others that are independent are affecting X_2 and so on.

-In the data generating mechanism for the outcome, I understand that the exposures

that are truly causal (beta_i != 0 in equation 2, line 512) are drawn from a uniform

distribution. I would suggest to look into more informative mechanisms, for instance if

one of the groups of exposures is affecting Y.

For example, what seemed to be a determinant

of performance in DOI: 10.7554/eLife.80063 is the number of non-causal exposures in

those blocks of exposures that contain some causal ones. That is, if X_1 is causal and

correlated with X_2, these two were grouped together and X_2 counted as a false positive.

Would this be the case with your approach as well?

-Seeing that a difference with the mentioned PCA-based methods is the ability to

provide inference on each individual exposure (rather than groups of exposures), I

believe a more thorough discussion of what exactly the coef() function in the plsr

pipeline is achieving. It was not clear how exactly there is a backtransformation

after the dimensionality reduction which PLS entails.

Minor comments

Table 1: Is the outcome scaled?

line 576: 'minizine' minimising

Fig.2, Table 1: 'ML-PL' MR-PL in legends

Reviewer #2: The authors propose a one-sample multivariable Mendelian randomization method called MR-PL. As a 2SLS method, it uses partial least square in the first stage to predict exposures using genome-wide significant SNPs, and then uses Lasso in the second stage to select and estimate potential causal exposures with non-zero coefficients, and incorporates de-sparsified Lasso for statistical inference. A heuristic approach for correcting winner's curse is also proposed. The authors apply the proposed method to infer the causal relationships between 37 white matter tracts and 180 traits using UKB individual-level data, and reveal some putative causal white matter tracts on smoking, blood vascular function-related traits, and eating behaviors. Overall, the manuscript is well-organized and generally straightforward. but I do have several concerns and questions:

Major:

1. The use of PLS in stage 1 by MR-PL was highlighted in the simulation and methods using OLS in stage 1 such as multi-2sls in general had larger MSE and type-I error. The authors explained in page 8 line 177 that 'PLS can better incorporate the correlation between exposures into the estimation of causal effect.' This wasn't clear to me as PLS was used in stage 1 and how is the correlation among exposures incorprated in the second stage. Furthermore, in the implementation of stage 1 in multi-2sls (and uni-2sls), was the whole matrix G used to predict all exposures (X), so for each exposure, some irrelavant/non-significant SNPs were used as the predictors as well? This may lead to bad prediction in stage 1 given a large number of SNPs included. Could the authors try he usual implementation, where for each exposure, only the relevant set of IVs are used for prediction and obtain X_hat. It was discussed in a previous multivariable 2sls called MV-IWAS [1] and it'd be better to discuss/compare with in this manuscript. So it seems to me the the main advantage of using PLS in stage 1 compared with the current implementation of OLS is that it can serve as a variable selection/dimension reduction purpose when predicting X.

2. The correction for winner's curse seems to be very heuristic and the current presentation is not convincing enough to me. Though the simulation showed improvement for estimation/MSE after WC correction, this may simply as the result of using more strong IVs and less weak IVs in the ideal scenario with no invalid IVs.

(1) Could the authors clarify page 31 lines 665-666 'Accordingly, it is expected that the sacrifice of instruments will lead to a larger reduction of MSE.'

(2) For the selection of c, is the intuition behind the highest index (decrease in MSE)/(decrease in IV) that MSE is reduced more (so the nominator is larger) and the less IVs are removed (so the denominator is smaller)?

(3) Furthermore, the parameter c is determined based on the simulation studies and fixed to be 20 in their real data analysis. This seemed somewhat arbitrary to me since c=20 was selected based on the *average* across all simulation scenarios, which may not be representative of the real data. More importantly, how would c be selected in other real data analysis for users? The authors should give detailed recommendation on this instead of simply fixing them to 20 as it is affected by sample size, p-value threshold based on their Eq.(10)

(4) After determining c, in their WC procedure, they directly subtracted a term from the original abs(z-score), then transformed back to the corrected p-value based on the normal distribution. My concern is that after this heuristic transformation does the score still follow the standard normal distribution?

(5) In page 12 line 251-252, 'For type I error control, WC-correction still enhanced the power of MR-PL,' Where is the power result?

3. The method itself cannot deal with pleiotropy which is the main concern in MR, and the Sargan's test is proposed to detect invalid IVs. Could the authors investigate how well the method perform in the presence of pleiotropic IVs and how well is the Sargan's test in this context?

4. The F statistics was used to evaluate the IV strength for each exposure, however, in the multivariable context, a conditional F statistics is also suggested to use for evaluating weak IV issue (see [2]). The authors should also report the conditional F-statistics.

5. In the abstract, the authors stated 'MR is more likely to be used in a single large dataset...' It gave readers an impression that one-sample MR is more common. However, I believe two-sample MR is the more popular and convenient one in current practice. It'd better to rephrase this sentence. Furthermore, in page 4 line 95, the ref [16] can account for overlapping samples, but it is limited to a small number of exposures.

6. In the real data analysis,

(1) Table S6 WC column only has ~21 categories while the analysis has 36 exposures, could the authors clarify this mismatch? Also, what does 'Average' in the WC column mean?

(2) In Table 1, I noticed that only Eosinophill percentage had two putatively causal IDPs, while other traits only had one. Could the authors comment on this? In particular, it would be interesting to compare the results between MR-PL and the univariable analysis. By comparing Table S12 and S8, I found that most of the significant IDP-trait pair identified by MR-PL were not significant at all by univariable analysis (and other methods). Is this expected or something going wrong here?

Minor:

1. Fig 2. row 4 column 1, the red dot didn't have error bar. And what is the 10-fold variance of MSE?

2. page 19 line 384, could you clarify in 0.05/34/69, what is 34 and what is 69?

3. page 27 line 575 'SNP I' should be 'SNP i'.

4. page 30 line 637, should it be 'N-'vector residual errors?

5. Could the authors clarify the leave-one-out analysis discussed in page 35 line 758 and page 19 line 382?

Ref:

[1] Knutson, K. A., Deng, Y., & Pan, W. (2020). Implicating causal brain imaging endophenotypes in Alzheimer’s disease using multivariable IWAS and GWAS summary data. NeuroImage, 223, 117347.

[2] Eleanor Sanderson, Wes Spiller, and Jack Bowden. Testing and correcting for weak and pleiotropic instruments in two-sample multivariable mendelian randomization. Statistics in medicine, 40(25):5434–5452, 2021.

Reviewer #3: Here the authors propose a new application for multivariable MR.

The method is very similar to 2SLS but uses Partial least squares for the first step and LASSO for the second step.

More over the author propose a correction for winners curse to be used in the one sample setting.

Overall the paper applies mostly well know statistical techniques so there is not much to be said from the theoretical point of view.

I have a slight issue with the winners curse correction as it doesn't seem to rely on a model apart to making the coefficient inversely proportional to sample size and -log(p). There is no simulation or any proof that this is actually proper through specific simulations so I am left unsure.

The method itself seems to be a bit uncalibrated as Type I error is much lower than expected. This I think is reflected by the real data application where out of 4420 tests only 26 have a p<0.05 instead of the expected >200.

So probably the test is underpowered. I realise that the estimates seem to be more precise when point estimates are compared to the original ones (lower MSE), since p-values are used these seem larger that expected, so in this respect it is hard to see the advantage of this method. Proper simulated power analysis should be presented.

In this context it is hard to evaluate the claimed results as in the end they could be the result of chance.

Linking brain areas to behaviour will basically always make sense, but other less. For example why would a specific brain area influence the width of platelets?

I would try to at least find some positive controls to see if everything is working fine.

For example Lipids, BMI, smoking and alcohol as exposures and CVD as outcome.

So overall although I see the potential advantages of the methods I am not convinced by the presented data that it is developed enough and the above issues should be addressed.

**Have all data underlying the figures and results presented in the manuscript been provided?**

Reviewer #1: Yes

Reviewer #2: Yes

Reviewer #3: Yes

PLOS authors have the option to publish the peer review history of their article (what does this mean?). If published, this will include your full peer review and any attached files.

Reviewer #1: **Yes: **Vasilis Karageorgiou

Reviewer #2: No

Reviewer #3: No

---

## [Decision Letter · Decision Letter 1]

24 Oct 2023

Dear Dr Zhao,

Thank you very much for submitting your Research Article entitled 'An augmented Mendelian randomization approach provides causality of brain imaging features on complex traits in a single biobank-scale dataset' to PLOS Genetics.

The manuscript was fully evaluated at the editorial level and by independent peer reviewers. The reviewers appreciated the attention to an important topic but identified some concerns that we ask you address in a revised manuscript.

We therefore ask you to modify the manuscript according to the review recommendations. Your revisions should address the specific points made by each reviewer.

Yours sincerely,

Zoltán Kutalik, PhD

Academic Editor

PLOS Genetics

Hua Tang

Section Editor

PLOS Genetics

Reviewer's Responses to Questions

**Comments to the Authors:**

Reviewer #1: It was interesting reading the review, the authors have put in extensive work in updating it.

I would add that it's more of the strenfth of association, that ia the magbitude of the parameter chosen in the sim's, thay guides the instrument strength, a stronger predictor of instr strength than number. Still i agree with the conclusion of the paragraph 297-307.

Appreciated the clarification on other types of more modifiable exposures and the discussion on brain tissue dynamics.

L195-199: If you could add a specification of the exact data generating mechanism for clarity purposes that would be useful

How is correlqtion achieved? Due to genetics?

The term β-map is a bit unclear.  Ι understand you refer to the many-element collection of the IDP-outcome effect estimates? You can define it in its first occurence.

Lines 563+: i still find it confusing how exactly the developped WC correction method works. It could be of use if the authors added a couple of sentences that summarise this part of the workflow in plain language. Especially how the end-output is apparently more precise; I would imagine that the SEs of the fasely selected IVs would be inflated afterwards.

Lines 684+: this passage spells out the difference of pca and pls in mr

Reviewer #2: Thank you for the revisions made to this paper, most of my comments have been resolved. However, I still have some remaining concerns.

Major:

1. In the ideal scenario simulation, univariable MR had inflated type-I error was expected, but for other multivariable methods, even 20 exposures are all included, it still had inflated type-I error. Did the authors explore why other multivariable MR methods and for some scenarios MR-PL had inflated type-I error even in the ideal scenario without pleiotropy and independent IVs (so no pleiotropy through LD as well)? Relatedly, how did the author define a type-I error? For example, for 20 exposures, 5 of them are non-causal, then it is counted a type-I error as long as one of the 5 p-values is smaller than 0.05? Similarly, how did the author define power when you have 15 causal exposures?

2. While I appreciate the authors made a huge effort to revise the part of winner's curse correction, it is still not convincing enough for me.

First, the intuition behind 'Select c with the lowest prediction R2 of outcome in MR from the grid search range.' was not clear. It seems to me the negative correlation between R^2 and reduction in MSE simply because when R^2 is large, correct for winner's curse or not didn't matter a lot, or the original method performed well enough in some sense. Could the authors verify this? S25 Figure was confusing - was the difference in R^2 and reduction in MSE driven by different c? Or the difference in R^2 was driven by different simulation setup? Then I am confused by how to select c based on R^2. Moreover, did the author check their suggested solution on real data?

Second, for the authors' response 2.2.4, transformation between P-values and Z-statistics is valid given the usual Z-score=beta_hat/se_hat which (asymptotically) follows standard normal under null hypothesis. However, my point is that when the original Z-score is transformed based on the proposed method, I don't think it still follows the standard normal that the authors use to calculate the corrected p-value.

3. In the calculation of the conditional F-statistics, could the authors clarify how they calculated for MR-PL? Step 1 says each exposure is regressed on the whole genotype matrix, but I assume in MR-PL the Xhat is calculated based on the PLS? In the real data analysis, all condition F-statistics are smaller than 2, are these calculated based on OLS or PLS?

Minor:

1. I'm not sure why a 95% line is plotted in the power plot. p<0.05 only tells you control type-I error at 5%.

2. In the real data analysis, when performing GWAS on IDP, why the authors also adjusted for BMI? How would the results be different if not adjusting for BMI?

Reviewer #3: I have no further comments to the authors. I think they did not get the suggestion of testing BMI LDL and HDL as exposures as outcomes and CVD as exposure to have a positive control, but I don't want to push the subject further.

**Have all data underlying the figures and results presented in the manuscript been provided?**

Reviewer #1: Yes

Reviewer #2: Yes

Reviewer #3: Yes

PLOS authors have the option to publish the peer review history of their article (what does this mean?). If published, this will include your full peer review and any attached files.

Reviewer #1: **Yes: **Vasilis Karageorgiou

Reviewer #2: No

Reviewer #3: No

---

## [Decision Letter · Decision Letter 2]

12 Dec 2023

Dear Dr Zhao,

We are pleased to inform you that your manuscript entitled "An augmented Mendelian randomization approach provides causality of brain imaging features on complex traits in a single biobank-scale dataset" has been editorially accepted for publication in PLOS Genetics. Congratulations!

Yours sincerely,

Zoltán Kutalik, PhD

Academic Editor

PLOS Genetics

Hua Tang

Section Editor

PLOS Genetics

Comments from the reviewers (if applicable):

Reviewer's Responses to Questions

**Comments to the Authors:**

Reviewer #1: The authors have responded appropriately

Reviewer #2: Thank you for putting effort in improving the work. I have no further comments.

**Have all data underlying the figures and results presented in the manuscript been provided?**

Reviewer #1: Yes

Reviewer #2: None

PLOS authors have the option to publish the peer review history of their article (what does this mean?). If published, this will include your full peer review and any attached files.

Reviewer #1: **Yes: **Vasileios Karageorgiou

Reviewer #2: No

**Data Deposition**

http://datadryad.org/submit?journalID=pgenetics&manu=PGENETICS-D-23-00635R2

**Press Queries**

---

## [Editor Report · Acceptance letter]

21 Dec 2023

PGENETICS-D-23-00635R2 

An augmented Mendelian randomization approach provides causality of brain imaging features on complex traits in a single biobank-scale dataset 

Dear Dr Zhao, 

We are pleased to inform you that your manuscript entitled "An augmented Mendelian randomization approach provides causality of brain imaging features on complex traits in a single biobank-scale dataset" has been formally accepted for publication in PLOS Genetics! Your manuscript is now with our production department and you will be notified of the publication date in due course.

With kind regards,

Lilla Horvath

PLOS Genetics

On behalf of:
